# Dynamic constriction and fission of endoplasmic reticulum membranes by reticulon

Javier Espadas[1,10], Diana Pendin [2,9,10], Rebeca Bocanegra[3,10], Artur Escalada [1], Giulia Misticoni[2], Tatiana Trevisan[2], Ariana Velasco del Olmo[1], Aldo Montagna[2], Sergio Bova[4], Borja Ibarra[3,5], Peter I. Kuzmin [6], Pavel V. Bashkirov[7], Anna V. Shnyrova[1], Vadim A. Frolov[1,8*] & Andrea Daga [2*]

The endoplasmic reticulum (ER) is a continuous cell-wide membrane network. Network formation has been associated with proteins producing membrane curvature and fusion, such as reticulons and atlastin. Regulated network fragmentation, occurring in different physio-logical contexts, is less understood. Here we find that the ER has an embedded fragmentation mechanism based upon the ability of reticulon to produce fission of elongating network branches. In *Drosophila*, Rtnl1-facilitated fission is counterbalanced by atlastin-driven fusion, with the prevalence of Rtnl1 leading to ER fragmentation. Ectopic expression of *Drosophila* reticulon in COS-7 cells reveals individual fission events in dynamic ER tubules. Consistently, in vitro analyses show that reticulon produces velocity-dependent constriction of lipid nanotubes leading to stochastic fission via a hemifission mechanism. Fission occurs at elongation rates and pulling force ranges intrinsic to the ER, thus suggesting a principle whereby the dynamic balance between fusion and fission controlling organelle morphology depends on membrane motility.

[1] Biofisika Institute (CSIC, UPV/EHU) and Department of Biochemistry and Molecular, Biology, University of the Basque Country, Leioa 48940, Spain.
[2] Scientific Institute, IRCCS E. Medea, Laboratory of Molecular Biology, Bosisio Parini, Lecco, Italy. [3] IMDEA Nanociencia, C/Faraday 9, Ciudad Universitaria de Cantoblanco, 28049 Madrid, Spain. [4] Department of Pharmaceutical and Pharmacological Sciences, University of Padova, Padova, Italy.
[5] Nanobiotecnología (IMDEA-Nanociencia) Unidad Asociada al Centro Nacional de Biotecnologia (CSIC), 28049 Madrid, Spain. [6] A.N. Frumkin Institute of Physical Chemistry and Electrochemistry, Russian Academy of Sciences, Moscow 119071, Russia. [7] Federal Research and Clinical Centre of Physical-Chemical Medicine, Moscow 119435, Russia. [8] IKERBASQUE, Basque Foundation for Science, Bilbao 48013, Spain. [9] Present address: Neuroscience Institute, Italian National Research Council (CNR), Padova, Italy. [10] These authors contributed equally: Javier Espadas, Diana Pendin, Rebeca Bocanegra
*email: vadim.frolov@ehu.eus; andrea.daga@gmail.com

The endoplasmic reticulum (ER) comprises two uninterrupted domains, the nuclear envelope and the peripheral ER. The peripheral ER is composed of structural elements with different membrane curvature and topology, from flat sheets and reticulated tubules to complex fenestrated structures. These elements are distributed throughout the cytoplasm of the eukaryotic cell as a membrane network enclosing a single lumen[1–3]. Network maintenance requires homotypic membrane fusion mediated by the atlastin family of dynamin-related GTPases[4,5]. Suppression of atlastin fusogenic activity leads to ER fragmentation[4], thus revealing an endogenous mechanism aimed at the reduction of ER connectedness. The existence of this mechanism has been confirmed by several reports showing ER disassembly during mitosis[6–8], reversible fragmentation of the ER both in neurons[9] and other cell types[10,11], and fragmentation of the ER prior to autophagic degradation[12,13]. Furthermore, fission of individual ER branches was recently detected by using super-resolution live-cell imaging of the ER network[14]. While no dedicated molecular machinery has been linked to ER fragmentation, few experimental observations suggest an involvement of reticulons[12,13,15], highly conserved integral ER membrane proteins implicated in shaping and stabilizing the tubular ER[16–19]. Notably, mutations in both Reticulon-2 and Atlastin-1 have been linked to the neurodegenerative disorder hereditary spastic paraplegia[20,21], corroborating their participation in coordinated functional and pathological pathways.

Overexpression of members of the Yop1 and reticulon families of proteins has been reported to cause severe constriction of ER branches[16,22] and ER fragmentation[15]. Fragmentation could proceed via the breakage of ER tubules, implicating high local curvature stress and membrane fission. Fragmentation was also linked to the shedding of small vesicles[15], a process whose significance in ER fragmentation, however, is not understood. Tubule fission would naturally antagonize the fusogenic activity of atlastin in the ER, making fusion/fission balance a paradigm in intracellular organelle maintenance. Despite the reported association between reticulons and ER fragmentation, direct involvement of reticulons has not been shown and the mechanism(s) of fragmentation remains obscure. Furthermore, creation of membrane curvature by reticulons was mechanistically linked to construction, not fragmentation of the tubular ER network, both in vitro and in vivo[5,16,23]. In agreement with involvement in formation rather than fragmentation of the tubular ER, purified reticulons reconstituted into lipid vesicles induced membrane curvatures insufficient to produce membrane fission[16,24].

Here, we reveal the mechanism underlying reticulon membrane activity that unifies these seemingly contradictory observations. We find that Drosophila Reticulon (Rtnl1), while promoting ER tubulation and enhancing the total curvature of ER membranes, is also responsible for ER fragmentation via membrane fission. Fragmentation occurs both at endogenous levels of Rtnl1, when unchallenged due to the absence of atlastin, and upon Rtnl1 overexpression. Corroborating these in vivo results, purified Rtnl1 reconstituted into dynamic lipid nanotubes produces curvatures ranging from moderate, as reported earlier[16], to those causing spontaneous membrane fission. In vivo, this ability of Rtnl1 to induce membrane fission is counterbalanced by atlastin, with the interplay between these proteins exerting the core control on total curvature and connectedness of the ER network in a living organism.

## Results

### Rtnl1 and atlastin display antagonistic genetic interaction.
Drosophila is a convenient model for studying Reticulon and its interaction with atlastin in vivo because its genome contains a single functional Reticulon gene (Rtnl1) and a single atlastin gene (atl). Homozygous Rtnl1[1] null flies[25] are viable and normal, while homozygous atl[2] null individuals[26] die at the pupa stage with a 2% rate of escapers. Surprisingly, we found that combining these two null mutations in homozygosity resulted in 84% adult survival (Fig. 1a). Hence, removal of Rtnl1 substantially alleviates the lethality associated with depletion of atlastin, indicating that a robust antagonistic genetic interaction between atlastin and Rtnl exists in Drosophila. This interaction was confirmed in the fly eye, where RNAi-mediated loss of Rtnl1 in an eye overexpressing wild-type atlastin resulted in increased severity of the atlastin-dependent small eye phenotype (Supplementary Fig. 1a), and in the nervous system, where the lethality produced by D42–Gal4-driven over-expression of atlastin in motor neurons is markedly enhanced in the Rtnl1[1] mutant background. EM tomography-based 3D reconstruction of the ER network in atl[2] neurons showed disconnected ER elements (Fig. 1b, Movies 1 and 2), supporting earlier observations of ER fragmentation after loss of atlastin[4]. Remarkably, depletion of Rtnl1 in the atl[2] null background restored a normal ER structure: ER network organization in Rtnl1[1];atl[2] neurons resembles closely that of control neurons comprising interconnected tubular and sheet-like elements (Fig. 1b, Movies 3–6). The observation that removal of Rtnl1 in the atl[2] null background restores both viability and ER shape strongly indicates that Rtnl1 is the force driving the morphological alterations and fragmentation of the ER caused by loss of the fusogenic activity of atlastin. Importantly, our data demonstrate that balancing the activities of atlastin and Rtnl1 is critical not only for the maintenance of ER architecture but also for organism survival.

### Rtnl1/atlastin ratio controls connectedness of the ER lumen.
The effects of reticulons on ER morphology have been linked to their ability to bend membranes: reticulons convert flat ER sheets into curved tubular structures, a phenomenon often referred to as tubulation[17,27]. In agreement with this notion, EM tomography in Rtnl1[1] mutant neurons revealed elongated unbranched ER sheets (Fig. 1b, Movies 7 and 8). This effect can be quantified using a simple metrics: the average length of ER profiles, corresponding to a cut through sheet-like structures, on thin EM sections (Fig. 1c). When compared with controls, Rtnl1[1] mutant neurons displayed elongated ER profiles, as previously reported for a different cell type[28]. Notably, ER profile elongation in Rtnl1[1] flies was suppressed by re-expression of transgenic Rtnl1 (Supplementary Fig. 1b, c) and Rtnl1 overexpression in a wild-type background led to profile length reduction (Fig. 1c, d). Crucially, EM analyses further revealed that this length reduction effect is normally compensated by the activity of atlastin. The length was substantially shortened in neurons lacking atlastin[4] thus demonstrating that loss of atlastin or Rtnl1 change profile length in the opposite direction. Consistent with this observation, the profile length reduction by Rtnl1 overexpression was exacerbated by simultaneous downregulation of atlastin. UAS-atl-RNAi,UAS-Rtnl1/tub-Gal4 neurons showed a significant decrease of average ER profile length when compared to UAS-Rtnl1/tub-Gal4, where endogenous atlastin can actively oppose Rtnl1 function, as well as to UAS-atl-RNAi/tub-Gal4 alone (Fig. 1c, d) where profile length reduction is due to uncontested endogenous Rtnl1. Even more striking than profile length decrease was the paucity of ER observed in UAS-atl-RNAi,UAS-Rtnl1/tub-Gal4 neurons (Fig. 1c), indicating that much of the network was broken up in small, unidentifiable fragments thus making our quantitative analysis biased towards visible, longer profiles. Finally, ER profile length in Rtnl1[1]/atl[2] double mutant neurons was comparable to that of control neurons demonstrating reciprocal compensation of the mutant phenotypes (Fig. 1c, d). These results demonstrate

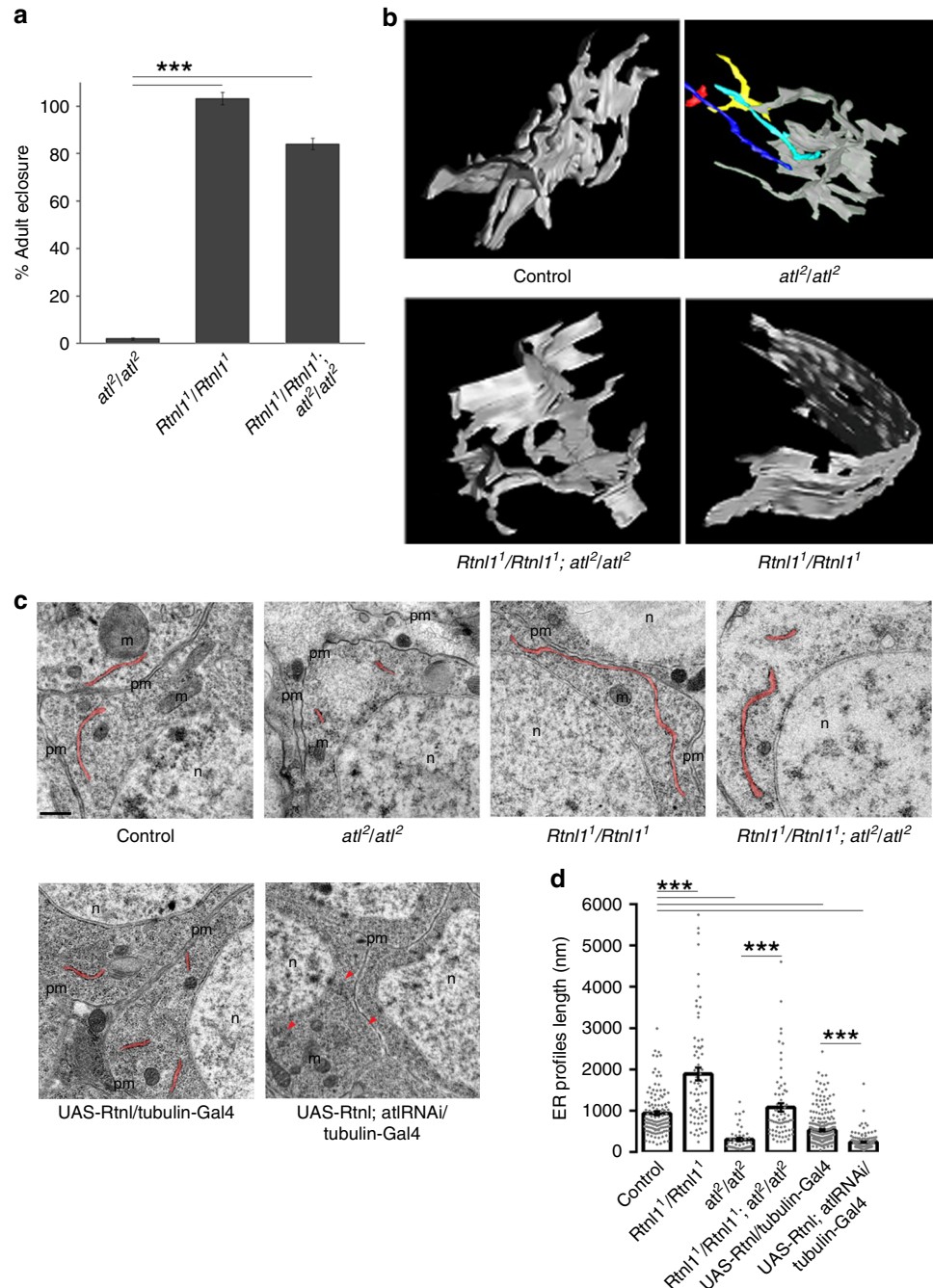

**Fig. 1** The genetic antagonism between *Rtnl1* and *atlastin* in *Drosophila* is reflected in morphological alterations of the ER. **a** The histogram displays the percentage of surviving adults, expressed as the ratio of observed over expected individuals, for the indicated genotypes. $n = 3$ independent experiments, statistical significance: unpaired two-tailed *t* test, ***$p < 0.001$. Error bars represent SEM. **b** EM tomography-based 3D reconstruction of portions of the ER network from neurons of the indicated genotypes. ER elements not connected are shown in color. Scale bar 200 nm. **c** Representative EM images of ventral ganglion neuronal bodies of the indicated genotypes highlighting ER profiles in red. Scale bar 0.5 μm. pm plasma membrane, m mitochondria, n nucleus. **d** average length of ER profiles measured on thin EM sections shown in (**c**), $n > 100$ ER profiles. Statistical significance: unpaired two-tailed *t* test, ***$p < 0.001$. Error bars represent SEM. Source data are provided as a Source Data file.

that atlastin counters the reduction of ER profile length mediated by either endogenous or transgenic Rtnl1 and that the atlastin/Rtnl1 ratio controls ER profile length (Supplementary Fig 1d, e), a parameter that can thus be used as a measure of the functional antagonism between Rtnl1 and atlastin in vivo.

Next, we used fluorescence microscopy to link Rtnl1 over-expression to the ER fragmentation seen upon downregulation of atlastin[4]. STED fluorescence microscopy of whole larva brain showed that Rtnl1 overexpression in neurons caused relocation of

the luminal ER marker GFP-KDEL to bright punctae in the perinuclear region similar to that observed upon downregulation of *atlastin* (Fig. 2a).

Accumulation of the luminal marker in these punctae was evident from the analysis of the fluorescence intensity distribution over the cytoplasm (Fig. 2b). Similar bright structures also emerged in larva muscles both upon *atlastin* downregulation and overexpression of Rtnl1 (Fig. 2c). We showed earlier by fluorescence loss in photobleaching (FLIP) that appearance of

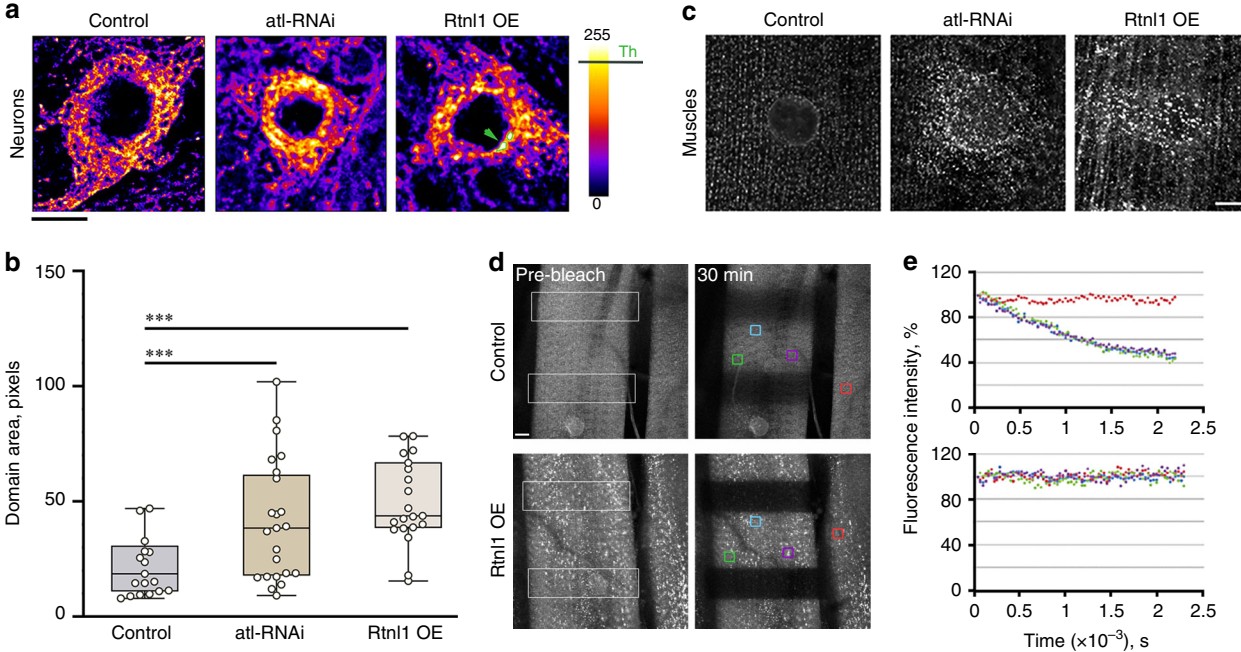

**Fig. 2** *Rtnl1* overexpression and loss of *atlastin* give rise to comparable defects in ER distribution and connectedness. **a** Deconvolved confocal STED projections showing comparable changes in the ER network appearance produced by *Rtnl1* overexpression (*Rtnl1* OE) and knockdown of *atlastin* (*atl*-RNAi) in *Drosophila* neurons labeled with the ER marker GFP-KDEL. The pseudo-color representation highlights emergence of bright fluorescent domains (examples marked by the arrow) in *Rtnl1* overexpression and *atl*-RNAi. Scale bar 5 μm. **b** *Rtnl1* overexpression and *atl*-RNAi show an increase of the total area of bright fluorescent domains (calculated using non-deconvolved STED projections with the brightness threshold Th = 200 indicated in (**a**)). Five randomly selected cells (total of 20 domains) were analyzed for each condition. Statistical significance: unpaired two-tailed *t* test, ***$p < 0.001$. Boxplots show IQR, whiskers indicate minimum and maximum of the dataset. **c** Emergence of bright fluorescent punctae in third instar larva muscle labeled by GFP-KDEL upon Rtnl1 overexpression or *atl*-RNAi. Scale bar 10 μm. **d** Representative images of FLIP performed by repetitive photobleaching of two regions (white outline box) in control and Rtnl1 overexpressing *Drosophila* larva muscles labeled with GFP-KDEL (left). Scale bar 10 μm. **e** Rates of fluorescence loss in four independent regions (color boxes) of control (top) and Rtnl1 overexpressing (bottom) muscle were quantified and graphed. The red box was chosen on an adjacent unbleached muscle as a control. Source data are provided as a Source Data file.

these punctae correlated with the fragmentation of the ER lumen, since in *atl*[2] muscles free diffusion of GFP-KDEL in the ER is abolished[4]. FLIP analysis of muscles ectopically co-expressing Rtnl1 and GFP-KDEL revealed a comparable loss of the diffusional exchange of GFP-KDEL between bleached and non-bleached ER regions (Fig. 2d). Therefore, when the atlastin/Rtnl1 ratio is decreased due to Rtnl1 overexpression or to loss of atlastin, the ER lumen becomes broken into disconnected fragments. Fragmentation puts a natural limit on the size of continuous ER elements thus providing a plausible explanation for the diminished length of ER profiles upon shifting atlastin/Rtnl1 balance towards the latter (Fig. 1d).

These results indicate that the functional antagonism between atlastin and Rtnl1 transpires not only in ER morphology changes but also in the overall connectedness of the ER network, with atlastin promoting fusion and Rtnl1 fragmentation of ER membranes. These complex interactions can be summarized by a simple kinetic model which assumes that atlastin and Rtnl1 cooperate in the production of tubular ER branches[15,19] but act antagonistically in regulating ER connectivity, with Rtnl1-driven ER fragmentation balancing the fusogenic activity of atlastin, as suggested above (Fig. 1b). The model shows that under these assumptions the ER profile length $L_p$ becomes directly proportional to the ratio of the atlastin and Rtnl1 concentrations ($L_p \sim$ [atl]/[Rtnl1], Supplementary Eq. (29)), justifying the use of $L_p$ as a metrics of the functional balance between atlastin and Rtnl1 in ER maintenance (Fig. 1d, Supplementary Fig. 1d, e) and highlights the pivotal role of Rtnl1-driven fragmentation in ER transformation and maintenance[15].

**Rtnl1 mediates constriction and fission of ER tubules**. To unravel the fragmentation mechanism, we resorted to ectopic expression of Rtnl1 in COS-7 cells whose outspread tubular ER network enables direct visualization and assessment of Rtnl1 activity. As in the fly, overexpression of Rtnl1 in COS-7 cells transformed the continuous ER network into bright punctae (Fig. 3a), with the extent of transformation being proportional to the amount of Rtnl1 in the cell (Supplementary Fig. 2a).

Quantitative analysis of the appearance of these fluorescent domains revealed that transformation of the ER became visible at ~12 h post transfection (12 h PT, Supplementary Fig. 2a, d) when the tubular network was still visible and dynamic. This transformation progressed to the fragmented state at 17–24 h PT, when the peripheral ER comprised mostly distinct subdomains whose lumen and membranes were visually unconnected (Fig. 3b, Supplementary Fig. 2c, d). Interestingly, GFP-Rtnl1 demonstrated significantly impaired fragmentation activity (Supplementary Fig. 2c, d) suggesting that it behaves as a partial loss of function mutant. To uncover the fragmentation pathway, we performed live imaging of ER dynamics in COS-7 at 12 h PT. Remarkably, we revealed scission of individual ER tubules both near the ends and in the middle portion of the tubules (Fig. 3c, red arrow, Movies 9–11), pointing to membrane fission as the mechanism underlying ER fragmentation. We scored a disconnection event as fission when it began as a localized constriction of a stable ER branch, followed by snapping of two disconnected parts of the branch in opposite directions (Fig. 3c, d). This pattern was distinct from transient tethering between ER branches (Fig. 3d). Importantly, fission always occurred in visibly

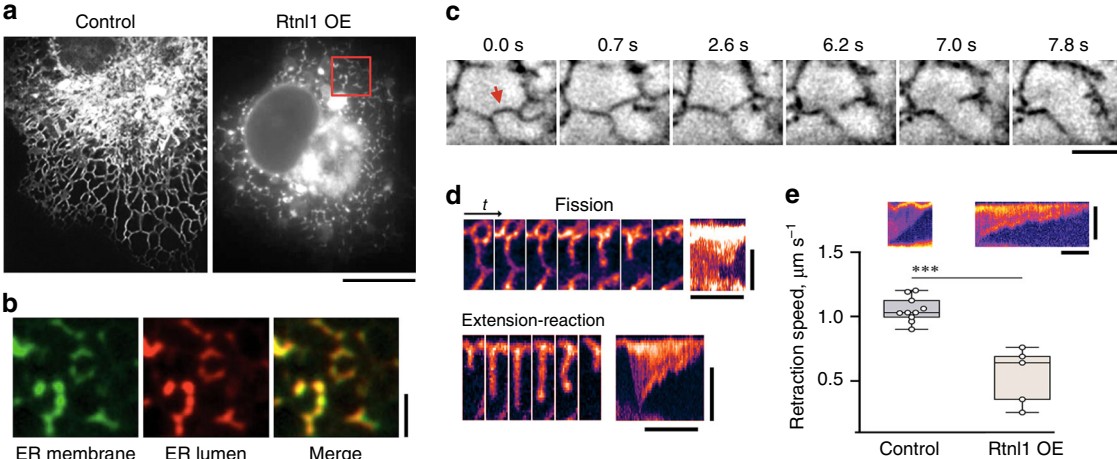

**Fig. 3** Altered dynamics and fission of ER branches during Rtnl1-driven fragmentation of tubular ER network. **a** Retraction and constriction of the ER network (labeled by mCHERRY-KDEL) in COS-7 cells expressing Rtnl1. Global ER constriction results in the appearance of multiple bright punctae of mCHERRY-KDEL fluorescence. Scale bar 10 µm. **b** Blow-ups of the ER in COS-7 cells co-transfected with Rtnl1-myc (identified by the simultaneous expression of nuclear CFP), mCHERRY-KDEL and GFP-Rtnl1 (24 h PT). Both fluorescence markers localize to visibly disconnected punctae. Scale bar 2 µm. **c** Image sequence showing scission (red arrow) of an ER branch in a Rtnl1-expressing COS-7 cell. Scale bar 2 µm. **d** Image sequences showing the scission (upper sequence, corresponding to that shown in (**c**)) of the ER branch correlated with the ring closure and the transient extension/contraction (lower sequence) of the ER branch. The sequences are followed by corresponding kymographs. The pseudo-color representation highlights the local constriction preceding the scission. Scale bars are 2 µm and 8 s. **e** Kymographs showing retraction of ER branches in control ($n = 10$ branches) and Rtnl1-expressing ($n = 5$ branches) cells. Scale bars are 2 µm and 2 s. The box-plot shows branch retraction speeds. Statistical significance: unpaired two-tailed $t$ test, ***$p <$ 0.001. Boxplots show IQR, whiskers indicate minimum and maximum of the dataset. Source data are provided as a Source Data file.

transforming parts of the ER network, indicating the involvement of axial forces and dynamic stresses (Fig. 3c, d; Movies 9 and 10). Such forces are intrinsic to actively remodeling, dynamic regions of the ER network[29], such as the peripheral ER where Rtnl1 driven fragmentation is the most apparent (Fig. 3a). However, Rtnl1 overexpression also caused significant slowing of the retraction of disconnected ER branches (Fig. 3e). Impaired retraction is consistent with stabilization of tubular ER branches by reticulons, the widely accepted function of reticulons. Curvature stabilization is, however, seemingly incompatible with the direct involvement of Rtnl1 in membrane fission.

**Rtnl1 constricts and stabilizes static membrane nanotubes.** To resolve the above contradiction, we reconstituted purified Rtnl1 into lipid nanotubes mimicking dynamic ER branches. We pulled the tubes from proteo-lipid bilayers formed on a surface of silica microbeads by proteo-liposome deposition (Supplementary Fig. 3c). Nanotube formation was monitored by fluorescence microscopy while the pulling force was measured by optical tweezers (Fig. 4a). Rtnl1 partitioning into the nanotube membrane was verified using fluorescently labeled protein (Fig. 4a, inset). Bulged and constricted regions appeared during pulling, dependently on the pulling speed ($v_t$) and Rtnl1 concentration in the reservoir (Fig. 4a). Quantification of the nanotube radius in the constricted regions ($R_t$) at $v_t = 0$ (Supplementary Fig. 4a–c) revealed that Rtnl1 creates static membrane curvature in the 0.1–0.3 nm$^{-1}$ range, proportional to the protein concentration (Fig. 4b). The nanotube curvature remains within the physiological range as the radii of 12–50 nm were reported for the ER tubules[16,30–32]. The highest curvature, measured at 1:150 Rtnl1/ lipid ratio, corresponds to that measured in cultured cells overexpressing reticulon[16]. Interestingly, similarly narrow ER tubules were recently revealed in neurons of both central and peripheral nervous system[30], likely indicating a tissue-specific regulatory mechanism.

While producing constriction, Rtnl1 alleviated the axial tensile force ($f_t$ Fig. 4a, Supplementary Fig. 5a–c), thus stabilizing the

nanotubes against retraction to the reservoir. This finding evokes reticulon-mediated inhibition of ER retraction upon prolonged microtubule depolymerization[33]. As $f_t R_t = 2\pi k_{eff} (1 - R_t J_s)$ (Supplementary Eq. (7)), the ~6-fold decrease of $f_t R_t$ (Fig. 4d) reflects two interrelated effects: the appearance of an intrinsic membrane curvature due to Rtnl1 in the reservoir ($J_s$) and the reduction of the effective bending rigidity of the nanotube membrane ($k_{eff}$) due to curvature-driven sorting of Rtnl1[34,35]. Ratiometric comparison of GFP-Rtnl1 fluorescence in the reservoir and constricted parts of the tubes showed significant sorting of the protein toward the nanotube[34] (Fig. 4c). For non-labeled Rtnl1 the sorting was detected as slow decrease of $f_t$ upon halting nanotube pulling (Supplementary Fig. 5a, b). Notably, the sorting efficiency recalculated from these force measurements (Supplementary Eqs. (9) and (10)) was higher than that obtained from the fluorescence microscopy observations on GFP-Rtnl1 (Fig. 4c), explaining the higher membrane constriction produced by the non-labeled protein (Supplementary Fig. 6). From combined fluorescence microscopy and force measurements datasets we further obtained $J_s = 0.02$ nm$^{-1}$, the intrinsic curvature of non-labeled Rtnl1 (the intrinsic membrane curvature at maximum protein incorporation) $J_p = 0.17$ nm$^{-1}$ and the membrane area occupied by the protein $a = 7.8$ nm$^2$ (Supplementary Eq. (10a)), comparable with those reported for Yop1p, another member of reticulon family[16]. High $J_p$ likely accounts for preferable outward orientation of Rtnl1 molecules in the proteo-liposomes (Supplementary Fig. 3c)[16], such asymmetric protein incorporation explaining considerable values of $J_s$ (Supplementary Eqs. (9) and (10a) and following discussion). Overall, our analyses of Rtnl1-containing nanotubes confirmed that Rtnl1 could operate as static curvature creator and stabilizer[36]. However, we revealed that curvature-driven sorting of Rtnl1 plays a significant role in the stabilization of membrane constriction by Rtnl1.

**Rtnl1 mediates nanotube fission via constriction-by-friction.** The weakened curvature activity of GFP-Rtnl1 in vitro provides a plausible explanation for its impairment of ER-fragmentation in

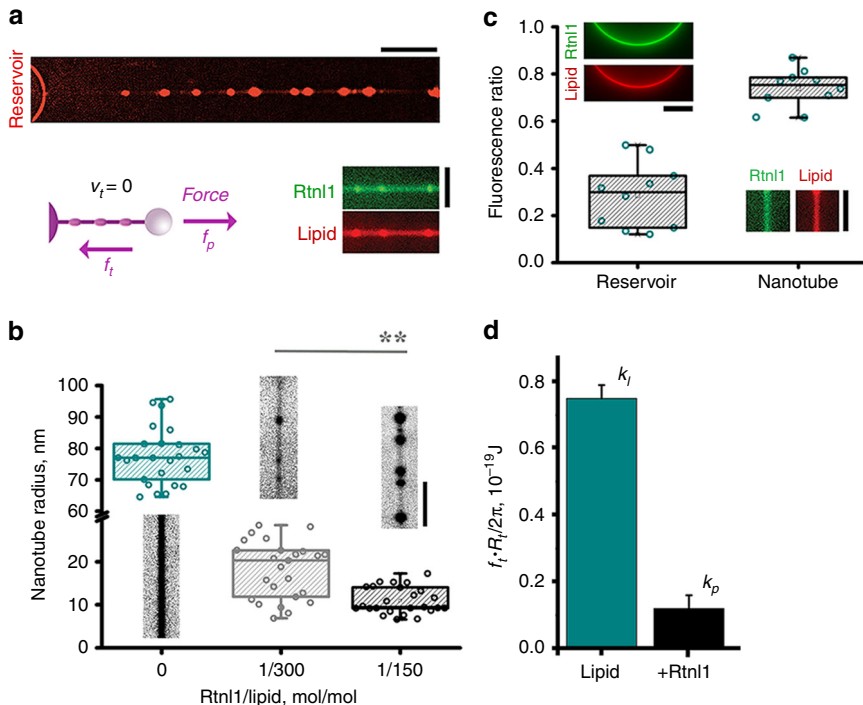

**Fig. 4** Static constriction of lipid nanotubes by Rtnl1. **a** Representative image of Rtnl1-constricted lipid nanotubes (Rh-DOPE fluorescence is shown) obtained with 1:150 Rtnl1/lipid (mol/mol). Scale bar 10 μm. The cartoon shows the static force balance $f_p = f_t$ where the pulling force ($f_p$) equals the tensile force ($f_t$). The insert shows incorporation of the Alexa488-Rtnl1 (green) into the nanotube (red) at 1:150 protein/lipid. Scale bar 2 μm. **b** Boxplots of the radii of the control nanotubes ($n = 25$ tubes) and the nanotubes constricted by Rtnl1 at 1:300 ($n = 25$ tubes) and 1:150 ($n = 25$ tubes) protein/lipid ratio; 3 independent Rtnl1 preparations were used. Representative images of the nanotubes are shown above/below the boxplots (Rh-DOPE fluorescence is seen, scale bar 2 μm). Boxplots depict IQR, whiskers indicate minimum and maximum of the dataset. **c** Differential incorporation of GFP-Rtnl1 into reservoir and nanotube membranes ($n = 8$ independent nanotube/reservoir pairs). The incorporation was measured as the ratio between the GFP-Rtnl1 and lipid (Rh-DOPE) fluorescence. Images show representative examples of the reservoir membrane and a constricted part of the nanotube membrane used for calculations (GFP and Rh-DOPE fluorescence, scale bars 4 μm). Statistical significance: unpaired two-tailed $t$ test, **$p < 0.01$. **d** Stabilization of constricted nanotubes by Rtnl1 (1:150 protein/lipid) measured as the decrease of $f_t R_t$. Mean $R_t$ and $f_t$ values are taken from panel (**b**) and Supplementary Fig. 5c, respectively: for pure lipid nanotubes $R_t = 77.4 \pm 1.8$ nm $n = 25$, $f_t = 6.9 \pm 0.4$ pN $n = 26$; for Rtnl1-containing nanotubes $R_t = 10.9 \pm 0.6$ nm $n = 25$, $f_t = 7.6 \pm 2.6$ pN $n = 10$, where $n$ is the number of independently formed nanotubes and errors represent SEM. Source data are provided as a Source Data file. Error bars represent SD ($n = 25$ tubes for the lipid column, $n = 10$ tubes for + Rtnl1 column).

COS-7 (Supplementary Fig. 2c, d). Yet neither purified GFP-Rtnl1 ($n = 26$) nor the wild-type protein ($n = 68$) could cause scission of static membrane nanotubes, as the maximal membrane curvature produced by either protein in vitro was insufficient to trigger nanotube destabilization and fission[24] (Fig. 4b, Supplementary Fig. 6b). Crucially, we found that pulling with constant speed ($v_t$) resulted in the surge of the nanotube curvature (Fig. 5a). The curvature increased linearly until reaching a plateau while a similar time pattern was recorded for the pulling force (Fig. 5a, Supplementary Fig. 5a). No increase in force or curvature was detected in control experiments where the membrane reservoir on the beads was made of lipid vesicles prepared the same way as their proteo–lipid counterparts (Fig. 5a, Supplementary Fig. 5a).

The synchronous increment of the curvature and force implied that growing membrane stresses cause nanotube scission (Fig. 5b, c; Movies 12 and 13)[37]. Fission was always detected in the nanotube regions near membrane reservoirs (Supplementary Fig. 3c), which are characterized by elevated constriction (Fig. 5b). As the increase of the nanotube length ΔL was limited, only a fraction of the tubes broke during the elongation (Supplementary Fig. 6a). Remarkably, the fission probability was significantly higher for Rtnl1 than for GFP-Rtnl1 at similar membrane concentration (Supplementary Fig. 6a), mimicking in vivo pattern (Supplementary Fig. 2c, d). A similar correlative impairment of in vitro and in vivo activities caused by a modification of the protein N-terminus was previously reported for Yop1p[16].

Pulling force dynamics during nanotube elongation and shortening (Fig. 5a, c, Supplementary Fig. 5a) evoked a behavior characteristic of viscous drag[37–40]. Accordingly, the increase of the axial force during pulling measured at the point of fission (Fig. 5c) showed characteristically weak logarithmic dependence on $v_t$[37,40] (Fig. 5d, black). Theoretical analyses revealed that such shear thinning[41] is due to intrinsic coupling between the curvature-driven Rtnl1 sorting toward the nanotube and tubule constriction during elongation (Supplementary Results, "Dynamic membrane constriction by Rtnl1"). By subtracting the lipid contribution to the force increment (Fig. 5d, cyan) and fitting the resulting $\Delta f(v_t)$ (Fig. 5d, inset) we found that Rtnl1 caused ~100-fold increase of the viscosity at 1:150 Rtnl1/lipid ratio. Notably, increasing Rtnl1:lipid ratio to 1:80 completely impaired nanotube production by pulling (12 out of 12 cases), likely, due to unsurmountable viscous resistance. The large membrane-inserting reticulon-homology domain, fully spanning the outer and partially the inner lipid leaflets, is likely to produce the viscous drag, further enhanced by reticulon oligomerization[33]. Interestingly, the theoretical analysis of Rtnl1 sorting suggests that GFP attachment impair protein oligomerization, thus diminishing the drag (Supplementary Information).

The relative increase of the nanotube curvature during pulling is not dramatic (about twofold, Fig. 5a, b). However, together with the static constriction produced by Rtnl1, it brings the curvature close to the stability limit (Fig. 5e, red). Critically, for

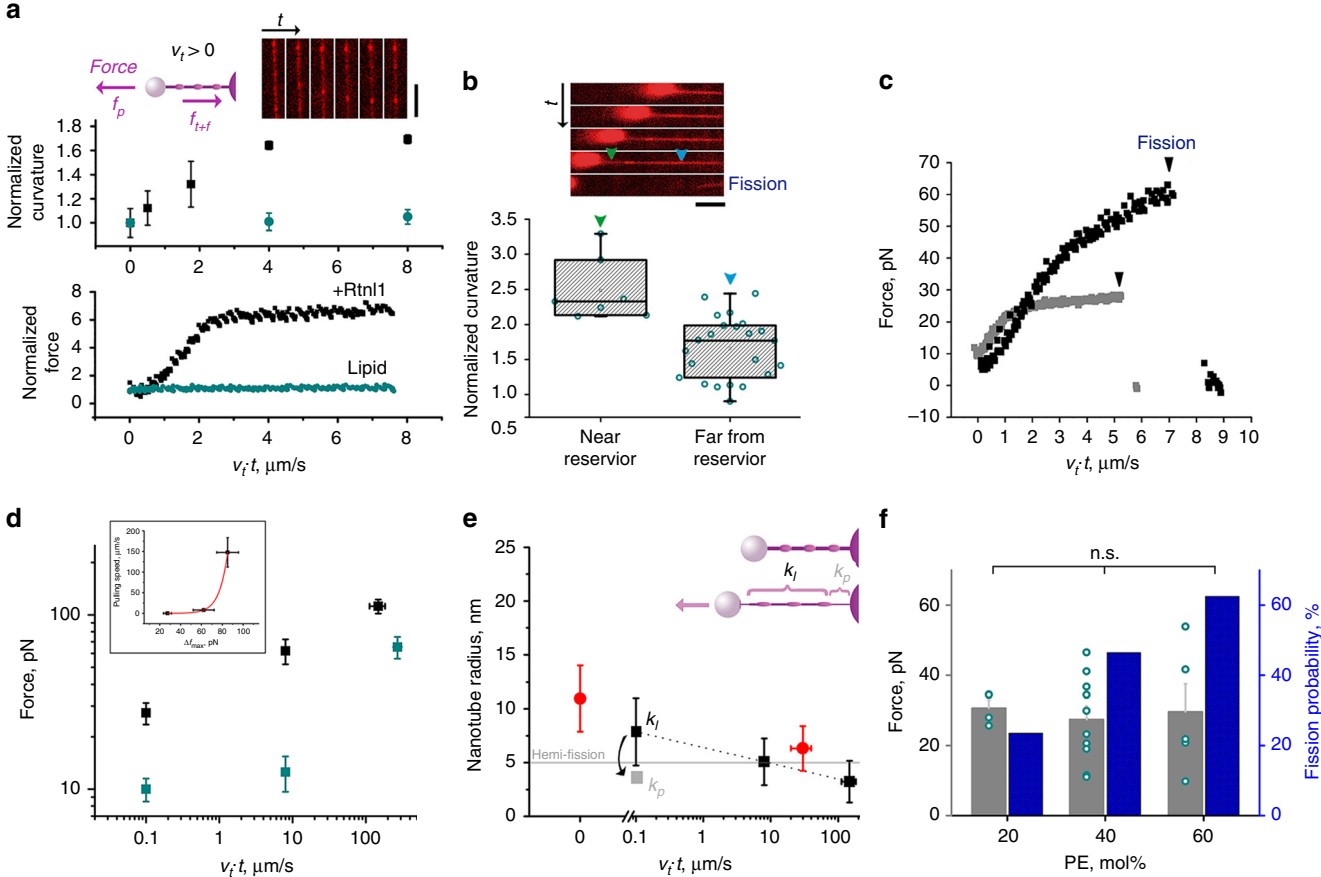

**Fig. 5** Constriction-by-friction mechanism of Rtnl1-driven membrane fission. **a** Simultaneous increase of the pulling force and the curvature (both normalized to their initial static values) during the elongation of control (cyan, $n = 3$ tubes) and Rtnl1-containing (black, $n = 3$ tubes) nanotubes at constant speed $v_t$. Error bars represent S.E.M. The image sequence shows the nanotube region used for the curvature calculations (Rh-DOPE fluorescence, scale bar 5 μm). The cartoon illustrates the dynamic force balance ($f_p = f_t + f_f$), where the pulling force is equal to the sum of the tensile and friction forces. **b** Frame sequence (100 ms/frame) showing scission (arrow) of Rtnl1-nanotube. Scale bar 2 μm. The histogram displays the increased membrane curvature near the membrane reservoir on the pulling pipette (green arrow, $n = 7$ tubes) as compared to the curvature far from the reservoir (blue arrow, $n = 20$ tubes). Boxplots show IQR, whiskers indicate minimum and maximum of the dataset. **c** The increase of the axial force during elongation of Rtnl1-nanotubes at 0.1 μm/s (gray) and 8 μm/s (black) speeds caused fission seen as an abrupt decrease of the force to zero (arrows). **d** Dependence of the axial force measured at the moment of fission of Rtnl1-containing tubes ($f_{Rtnl1}$, black, $n = 10$ ($v_t = 0.1$ μm/s), $n = 5$ ($v_t = 8$ μm/s) $n = 12$ ($v_t = 106$ μm/s)) or maximal force measured during 10 μm elongation of lipid tubes ($f_{lip}$, cyan, $n = 9$ ($v_t = 0.1$ μm/s), $n = 34$ ($v_t = 8$ μm/s) $n = 24$ ($v_t = 106$ μm/s)) on the elongation speed $v_t$. The inset shows the force difference $\Delta f_{max} = (f_{Rtnl1} - f_{lip})$ dependence on $v_t$, where the red line is the logarithmic fit (Supplementary Eq. (27)). The error bars show SD, three independent Rtnl1 preparations were used. **e** Radial constriction of the membrane nanotube measured by fluorescence microscopy (red, the error bars show SD, $n = 25$ tubes for both static and dynamic constriction) and recalculated from the force increase (shown in (**d**)) using either $k_l$ (black) or $k_p$ (gray). **f** Effect of the addition of conical lipids on the pulling force at the moment of fission and fission probability. The PE amounts indicated correspond to 20 mol% DOPE ($n = 17$ tubes, fission in 4 out of 17 cases) 40 mol% DOPE ($n = 28$ tubes, fission in 13 out of 28 cases) and 40 mol% DOPE + 20 mol% plasma ($n = 8$ tubes, fission in 5 out of 8 cases). Statistical significance: one-way ANOVA with multiple comparison. Source data are provided as a Source Data file.

this coupling between static and dynamic constriction, Rtnl1-constricted regions of the nanotube preserved elastic susceptibility to bending. The reduction of the radius ($R_t$) of the constricted regions during the initial "elastic" stage of pulling (when both force and curvature grow linearly with the extension length[41], Fig. 5a, c, Supplementary Fig. 5a) can be expressed as $\Delta R_t = 2\pi k_l \Delta\left(\frac{1}{f}\right)$, where $k_l$ is the bending rigidity modulus of the control lipid nanotube[37,41] (Supplementary Eq. (18)). $\Delta R_t$ calculated with $k_l$ and the fission force values from Fig. 5d (black), matched that directly measured by fluorescence microscopy (Fig. 5e, black and red). Hence, the nanotube regions pre-constricted by Rtnl1 retained lipid elasticity indicating sparse (~20% area, Supplementary Eqs. (9) and (10a)) Rtnl1 coverage of the nanotube[16].

Near the membrane reservoir friction-driven constriction is further enhanced by sorting of Rtnl1 toward the progressively thinning nanotube (Fig. 5b, e cartoon) as the sorting diminishes the effective bending rigidity of the Rtnl1-containing nanotubes (Figs. 4d, 5e, Supplementary Eq. (9)). In the ER network, this sorting effect would facilitate local constriction during slow elongation and enhance fission of the tubules pulled from low curvature portions of the network, such as ER sheets, explaining their prevalence at the late stages of Rtnl1 overexpression (Fig. 3a, Supplementary Fig. 2a, b). The dynamic, labile constriction of membrane tubules by Rtnl1 is strikingly different from that imposed by rigid protein scaffolds, such as Dynamin1 helix completely encaging lipid nanotubes[42]. Notably, Dynamin1 produced static constriction similar to Rtnl1, yet, Dynamin1 scaffolds retain their geometry during elongation

thereby preventing force-driven constriction and ensuing fission (Supplementary Fig. 4d, e).

**Rtnl1-driven fission proceeds through stochastic hemi-fission**. Despite the sparse, limited Rtnl1 coverage of the nanotube, the constriction-by-friction mechanism operates already at low, physiologically relevant elongation rates. The forces measured ranged from $27.4 \pm 3.9$ pN at $v_t = 0.1$ μm/s to $62.1 \pm 10.1$ pN $v_t = 8$ μm/s, comparable with the forces reported in nanotubes pulled from the ER and Golgi membrane networks (20–40 pN)[43]. At $v_t = 0.1$ μm/s 15 out of 32 tubes broke $32.4 \pm 2.6$ s after the beginning of elongation (Fig. 5c, gray curve), corresponding to an average 3.2 μm extension of the tube before fission. Hence, in vitro, Rtnl1 causes nanotube scission at elongations, speeds and forces normally present within the ER network. Notably, during such slow elongation the nanotube curvature approached but did not exceed the hemi-fission threshold (Fig. 5e). Furthermore, due to limited elongation length the curvature stress is applied only transiently during the elongation (for the nanotube elongation $\Delta L < 10$ μm the stress was applied for less than a second at $v_t = 8$ μm/s, see Fig. 5c). At such subcritical transient curvature stress the fission becomes a stochastic, thermal-driven process[44] as indicated by the observation that at $v_t = 8$ μm/s only 5 out 17 nanotubes broke (at $62.1 \pm 10.1$ pN) although unbroken tubes survived the same stress ($63.6 \pm 5.8$ pN). We also noted that a sustained increase in curvature stress could be prevented by incremental, short-step elongation (Supplementary Fig. 5d). The stochastic character of fission likely explains the low occurrence of fission events in the ER under normal circumstances when Rtnl1 concentration and pulling forces and speeds are within physiological limits.

Importantly, even significant force increases measured during high-speed pulling ($v_t$ above 100 μm/s) were insufficient to trigger fission of control lipid nanotubes[24]. High speed pulling produced similar force increase in Rtnl1-containing and lipid-only nanotubes (Fig. 5d Black and cyan) but it caused no scission in control tubes (in 21 out of 21 cases) while producing breakage of 12 out of 12 proteo-nanotubes. To directly compare the stress effect, we subjected lipid nanotubes to elevated forces (above 30 pN) for the same amount of time ($0.74 \pm 0.23$ s) as Rtnl1 nanotubes elongating at 8 μm/s. None of 15 lipid tubes broke as compared with 30% fission rate detected with the proteo-nanotube. It has been suggested that protein presence in the nanotube could facilitate the formation of a pore even under moderate tensile stress, thus leading to nanotube fission via membrane poration[37]. However, our data indicate that in Rtnl1-containing tubes force-driven constriction is critically enhanced by the intrinsic curvature and curvature-driven sorting of Rtnl1, bringing the nanotube curvature close to the hemi-fission threshold (Fig. 5e, red)[24] and thus biasing fission toward the pore-free remodeling path[44,45]. In support of this hypothesis we found that increasing the membrane concentration of conical-shaped lipids that are abundant in the ER[46] and are known inhibitors of pore formation as well as promoters of non-bilayer intermediates of hemi-fusion and hemifission[47], drastically increased the fission probability (Fig. 5f).

## Discussion

Ever since the discovery of homotypic fusion of ER membranes by atlastin there have been indications in the literature of the existence of an endogenous mechanism balancing unceasing fusion during ER network maintenance[48]. Recent studies, both in vitro and in vivo, reiterated the physiological importance of ER fragmentation and linked it to the curvature-creating proteins operating in the ER[15,49]. Yet, the puzzle remained as to how

proteins implicated in making the tubular ER network, such as reticulons, could also mediate fragmentation of the same network. Our results demonstrate that these seemingly opposite functions can indeed exist in a single protein, Rtnl1, combining two different modes of curvature creation, static, and dynamic. The static mode, associated with local membrane bending by the membrane-inserting domains of reticulons[48,49], accounts for mechanical stabilization of membrane tubes[16] (Fig. 4d). The dynamic mode, associated in this work with the increased viscosity of Rtnl1-containing membranes, accounts for friction-driven constriction of elongating membrane tubules, leading to their scission (Fig. 5b, c). Dynamic coupling between these two modes via curvature-driven sorting of Rtnl1 toward the nanotube is absolutely critical for fission to occur. Viscous drag alone would produce nanotube constriction only at elevated tensile stress and thus result in the mechanical rupture of the membrane[24]. Dynamic accumulation of Rtnl1 in the curved nanotubes, however, critically amplifies constriction so that scission can happen at reduced tensile stress, via a hemi-fission mechanism (Fig. 5e, f). Thus, the hemi-fission curvature threshold can be reached at physiological elongation, speeds and forces, within a range of Rtnl1 concentration that creates only the moderate static curvatures required for ER tubule stabilization[16] (Fig. 4b). Hence, in the dynamic ER network Rtnl1 readily combines its membrane curvature stabilization and fission activities without risking the leakage of the ER lumen contents into the cytoplasm.

In ER maintenance, membrane fission by Rtnl1 must be balanced by atlastin-mediated membrane fusion (Fig. 2a, b). Fundamentally, this balance is described by a kinetic model which explicitly accounts for the two opposing functions of Rtnl1, static curvature stabilization and dynamic fission (Supplementary Results, "Kinetic model of atlastin-Rtnl1 interactions in ER maintenance"). The intrinsic sensitivity to membrane dynamics suggests a paradigm of dynamic regulation of ER topology linking membrane fusion and fission with membrane motility. This paradigm implies that ER fragmentation, a process crucial in physiological conditions, for example maintenance of ER morphology and ER-phagy, and likely involved in neuropathological processes[50] can be implicitly controlled by multiple factors connected to ER motility and stresses, with Rtnl1 constituting the core element of the ER-specific membrane fission machinery.

## Methods

**Drosophila genetic and behavioral analysis**. Fly culture and transgenesis were performed using standard procedures. Rtnl1-PB cDNA was cloned in the pUAST vector for *Drosophila* transgenesis in frame with a HA tag.
    Primers used:
    Rtnl1-PB HA 5′-AGCTGAATTCATGTACCCATACGATGTTCCTGACTATGCGGGCTCCGCATTTGGTGAAACC-3′
    Rtnl1-PB 5′-AGCTTCTAGATTACTTGTCCTTCTCAGAC-3′
    Several transgenic lines for UAS-HA-Rtnl1 were generated and tested. *Drosophila* strains GMR-Gal4, D42-Gal4, tubulin-Gal4, arm-Gal4, pUASp:Lys-GFP-KDEL were obtained from the Bloomington Drosophila Stock Center. UAS-Rtnl1-RNAi lines were obtained from the Vienna *Drosophila* RNAi Center (v7866 and v33919). Lifespan experiments were performed with 200 animals for each genotype. Flies were collected 1 day after eclosion and placed in vials containing 50 animals. The animals were maintained at 25 °C, transferred to fresh medium every day, and the number of dead flies was counted. Lifespan experiments were repeated three times.

**Fluorescence loss in photobleaching (FLIP)**. FLIP experiments were performed as follows. Experimental larvae expressing UAS-GFP-KDEL were dissected in $Ca^{2+}$-free HL3 and analyzed using a Nikon C1 confocal microscope through a Nikon Fluor 60× water immersion objective. Two different region of interests (ROIs) along muscle 6 or 7 in the abdominal segment 4 were selected and bleached in 20 iterations, at 100% laser power, followed by three scanning images every 15 s. The bleaching protocols were repeated for 1 h[4]. The experiments were repeated at least three times. To create fluorescence recovery curves, fluorescence intensities were transformed into a 0–100% scale and were plotted using Excel software.

**Electron microscopy.** *Drosophila* brains were fixed in 4% paraformaldehyde and 2% glutaraldehyde, dehydrated, embedded in Epon and sectioned using conventional methods[4]. EM images were acquired under a FEI Tecnai-12 electron microscope. EM images of individual neurons for the measurement of the length of ER profiles were collected from three brains for each genotype. At least 20 neurons were analyzed for each genotype. Quantitative analyses were performed with ImageJ software[51].

**Electron tomography.** Epon-embedded *Drosophila* larval brains were cut transversally to the ventral nerve cord with Leica Ultracut UCT ultramicrotome. Totally, 200–250 nm thick serial sections were collected on formvar carbon-coated slot grids and 10 nm colloidal gold particles were deposited on both surfaces for fiducials. Samples were imaged on a FEI Tecnai G2 20 operating at 200 kV (Lab6) with a FEI eagle 2k CCD camera at a nominal magnification of 14,500 that resulted in a resolution of 1.5 nm per pixel. FEI single tilt tomography holder was tilted over a range of ±65° according to a Saxton's scheme (2° starting angle, for a total of 87 images collected) using the FEI Xplore3d acquisition software. Tilted images alignment, tomography reconstruction (WBP) and tomograms joining was done with the IMOD software package[52]. ER structures were rendered by manually segmenting the membranes of ER profiles using IMOD software[53].

**Super-resolution imaging.** Third instar larva brain neurons were imaged in fixed larvae preparations. Stacks of optical sections (300 nm apart) from the neurons were obtained in a Leica TCS STED CW SP8 super-resolution microscope with a $63 \times 1$, 40NA oil ($n = 1518$) objective using an Argon laser with an excitation line at 458 nm and depletion laser at 592 nm, adjusting the pinhole to one Airy unit, pixel size = 71 nm. Super-resolution images were deconvolved with PSF Generator and DeconvolutionLab plug-ins for ImageJ (Biomedical Imaging Group, École Polytechnique Féderale de Laussanne, big www.epfl.ch). The Born & Wolf 3D Optical Model generated a theoretical PSF using PSF Generator. Deconvolution was performed with the Richardson-Lucy algorithm using 70–80 iterations. Background subtraction was done before the deconvolution process. In each case, PSF stacks were generated with the same number of z-planes as of the image stacks deconvolved.

**Calculation of the bright puncta area in super-resolution images.** The super-resolution images of larval ventral ganglion neurons from control, Rntl1 OE and atl-RNAi flies were thresholded at 90% of maximal intensity (as shown in Fig. 3a). The resulting binary images were used as the masks defining the bright puncta in the images. The area and mean fluorescence intensity of the puncta were further analyzed using Analyze Particles algorithm of ImageJ[51].

**Cell culture and transfection.** mGFP-Rtnl1 and Rtnl1 were cloned in pcDNA3 for mammalian cell expression. We also generated a pcDNA3 plasmid containing two tandem CMV transcription units, one expressing a nuclear-CFP and the other expressing Rtnl1-myc.
Primers used:
For Rtnl1-PB 5′-AGCTGAATTCATGTCCGCATTTGGTGAAACC-3′
For mGFP 5′-AGCTGAATTCATGGTGAGCAAGGGCGAGGAGC-3′
Rev Rtnl1-PB 5′-AGCTTCTAGATTACTTGTCCTTCTCAGAC-3′
COS-7 cells (ATCC® CRL-1651™) were cultured in DMEM (HyClone, high glucose, from Thermo Scientific) supplemented with 10% fetal bovine serum and 50 μg/ml Gentamicin. For fluorescence microscopy experiments cells were plated in 35 mm low wall μ-Dishes (Ibidi GmbH) and transfected with vectors for expression of mCHERRY-KDEL (Clontech) and Rtnl1 (2 μg DNA each) using lipofectamine 2000 (Invitrogen) according to the manufacturer procedure.

**Live imaging in COS-7 cells.** Live-cell imaging was performed using a microscope stage top incubator INUG2 (Tokai Hit) equipped with an objective heater to maintain the cells at 37 °C with 5% $CO_2$. The incubator was installed on the stage of Olympus IX71 epi-fluorescence inverted microscope equipped with 150× 1.45NA TIRFM objective lens, iXon-EMCCD camera (Andor, Ireland), and BrightLine filter sets (Semrock, USA) for Alexa488 (485/524 nm excitation/emission), Rhodamine (543/593 nm excitation/emission) and CFP (434/479 nm excitation/emission). Image sequences were acquired using μManager open source software[54] at 10 or 30 fps. Images were further processed using ImageJ for cropping, background, and brightness/contrast adjustments[55]. To quantify the Rtnl1-driven constriction of the peripheral ER network in COS-7 cells still images extracted from the image sequences obtained at different times post-transfection were used. Two to three 30–40 μm² ROIs covering the peripheral ER (as in Fig. 3a) were used. All point histograms of the pixel fluorescence intensity were obtained for each ROI for the control COS-7 cells and cells expressing Rtnl1 and GFP-Rtnl1 (the corresponding probability densities are shown in Supplementary Fig. 2c).

**Protein expression and purification.** The cDNA encoding Rtnl1-PB isoform was obtained from the *Drosophila* Genomic Resource Center (LD14068). Rtnl1-PB cDNA was subcloned into the pQE-30 vector either alone or in frame with N-terminal mGFP to generate His-Rtnl1 and His-mGFP-Rtnl1.

Primers used:
For Rtnl1-PB 5′-AGCTGGATCCATGTCCGCATTTGGTGAAACC-3′
For mGFP 5′-AGCTGGATCCGTGAGCAAGGGCGAGGAGC-3′
Rev Rtnl1-PB 5′-AGCTAAGCTTTTACTTGTCCTTCTCAGAC-3′
His-Rtnl1 or His-mGFP-Rtnl1 were expressed in M15 bacteria. Bacteria were lysed in buffer containing 4% Triton X-100 and the resulting lysate was incubated with Ni-NTA Resin (Sigma-Aldrich) and washed sequentially with decreasing concentrations of Triton X-100 to a final 0.1%. His-Rtnl1-HA or His-mGFP-Rtnl1 were then eluted in buffer containing 0.1% Triton X-100 and used immediately or flash frozen in liquid nitrogen for storage at −80 °C. Wild-type human Dynamin 1 was produced in Sf9 insect cells and purified as follows[56]. Sf9 cells were transiently transfected with cDNA encoding Dynamin 1 subcloned in pIEx-6 vector (EMD Millipore, Billerica, MA) for protein production. The protein was purified by affinity chromatography using GST-tagged Amphiphysin-II SH3 domain as the affinity ligand. Purified Dynamin 1 was dialyzed overnight in 20 mM Hepes, 150 mM KCl, 1 mM EDTA, 1 mM DTT, and 10% (v:v) glycerol (pH 7.5), aliquoted, flash-frozen in liquid nitrogen, and stored at −80 °C. Protein concentration was determined with the BCA assay kit following the manufacturer's procedure (ThermoFischer Scientific, USA).

**Protein labeling.** Purified Rtnl1 was labeled with Alexa Fluor™ 488 maleimide (ThermoFischer Scientific, USA) according to the manufacturer's instructions. Dye excess was removed by using dye removal columns (ThermoFischer Scientific, USA). The labeling efficiency was assayed by absorption measurements and was ~0.2 dye/protein. Dyn1-Atto488 (generously provided by Dr. Sandra Schmid, UTSouthwestern) was used in some of the experiments.

Large unilamellar vesicles (LUVs) preparation. Dioleoyl-phosphatidyl-choline (DOPC), Dioleoyl-phosphatidyl-ethanolamine (DOPE), 1-(1Z-octadecenyl)-2-oleoyl-sn-glycero-3-phosphoethanolamine (C18(Plasm)-OPE), Dioleoyl-phosphatidyl-serine (DOPS), Rhodamine-DOPE (Rh-DOPE), cholesterol (chol), and phosphatidylinositol 4,5-bisphosphae ($PI4,5P_2$), all from Avanti Polar Lipids, were used to prepare the LUVs. Unless indicated otherwise, for Rtln1 reconstitution lipid mixtures of DOPC:DOPE:DOPS:Chol:Rh-DOPE at 39.5:40:10:10:0.5 mol% were used. In experiments shown in Fig. 5f, the DOPC:DOPE ratio in the mixture was changed to 59.5:20 mol% or to 19.5:40:20 DOPC:DOPE:C18(Plasm)-OPE (indicated as 60 mol% PE in Fig. 5f). In experiments with Dyn1, the lipid composition was DOPE:DOPC:Chol:DOPS:Rh-DOPE:$PI4,5P_2$ 39.5:38:10:10:0.5:2 mol%. The lipid stocks mixed in chloroform were dried under a stream of $N_2$ gas followed by further drying under vacuum for 120 min. The lipid films were resuspended in working buffer (20 mM HEPES pH 7.4, 150 mM KCl, 1 mM EDTA) to a final total lipid concentration of about 10 mM. For Dyn1 experiments, the film was resuspended in 1 mM Hepes. In both cases, LUVs were formed by 10 freeze–thaw cycles followed by extrusion through polycarbonate filters with 100 nm pore size (Avanti Polar Lipids, USA).

**Rtnl1 reconstitution into LUVs.** Preformed LUVs were diluted to 1 g/L (1.6 mM) with working buffer and titrated with Triton X100 to measure the optical density at 540 nm to find the optimum for LUV destabilization[57]. The detergent-destabilized liposomes (final concentration ~0.2 mM) were then mixed with Alexa488-His-Rtnl1-HA, His-Rtnl1-HA (referred to as Rtnl1) or His-GFP-Rtnl1 (referred to as GFP-Rtnl1) (11 μM in working buffer with 10% glycerol). After 15 min of co-incubation at RT with gentle agitation, the detergent was removed with BioBeads® SM-2 adsorbant (BioRad), added four times to the proteo-lipid mixture (at time 0, +30 min, +90 min, and +150 min), followed by ON incubation with the beads[58]. The sample was then centrifuged for 1 h at 15,000×g to remove the Bio-Beads® and non-incorporated protein. Rtnl1 incorporation into LUVs was measured by SDS-PAGE of the supernatant. Three independently prepared protein batches were used in the experiments. To compare the efficiency of Rtnl1 and GFP-Rtnl1 incorporation into proteo-LUVs (Supplementary Fig. 3b) we used a flotation assay based on a 3–6–9–12–15–20–30–40% OptiPrep™ density gradient. Optima MAX ultracentrifuge (Beckman Coulter) with MLS-50 rotor was used. Upon 2-h centrifugation at 4 °C, the fraction was collected from the bottom. The protein content of each fraction was analyzed by SDS-PAGE and the Rh-DOPE fluorescence of each fraction was analyzed by fluorescence spectroscopy using 96-well plate reader. No protein was detected in non-fluorescent fractions. Free proteins did not migrate to the proteo-LUVs parts of the density gradient.

**Formation of lipid and proteo-lipid reservoir membranes on silica and polystyrene beads.** Proteoliposomes were dialyzed against 1 mM Hepes and 1 mM trehalose solution. Ten microlitre of the freshly dialyzed proteo-liposome or LUVs in 1 mM Hepes solution were mixed with 2 μl of 40 μm silica or 5 μm polystyrene beads (Microspheres–Nanospheres, USA), deposited on a Teflon film in small drops and then dried in vacuum for 20 min. A 10 μL plastic pipette tip was cut from the bottom to approximately 2/3 of its original size. The cut tip was used to take 6 μL of 1 M TRH solution buffered with 1 mM Hepes. The tip was carefully detached from the micropipette, and a small portion of the beads covered by dried proteo-lipid or lipid films were picked up (using a fire closed patch glass capillary) and deposited into the TRH solution from the top of the tip. The tip was then carefully introduced into a home-made humidity chamber and subjected to

10–20 min incubation at 60 °C. Then the beads were transferred to the observation chamber filled with the working buffer, which was pretreated with bovine serum albumin to prevent lipid attachment to the glass[59]. Upon immersion in the working buffer, lipid or proteo-lipid film swelling was followed as spontaneous GUV or proteo-GUV formation. Proteo-GUVs attached to beads are shown in Fig. S5. Estimation of protein incorporation into the proteo-GUVs is described in Fig. S6. Optionally, by decreasing the membrane reservoir on the beads, the formation of vesicles was suppressed, and hydrated lamellas around the bead were formed instead.

**Pulling membrane nanotubes from the reservoir membrane**. Glass micropipettes were prepared with the P-1000 micropipette puller (Sutter Instruments, USA). A streptavidin covered polystyrene bead (2 μm in diameter, Microspheres-Nanospheres, USA) was trapped at the tip of the micropipette by suction and used to pull the tube from the lipid or proteolipid film or GUV whose membrane was previously doped with 0.2% of biotinylated-DOPS lipid (Avanti Lipids, USA). A micro-positioning system, based on 461xyz stage and high-resolution NanoPZ actuators (Newport, USA), was used for micropipette positioning, while calibrated piezo-actuator controlled by ESA-CXA μDrive three-axis controller (Newport, USA) was used for constant-speed nanotube elongation. Same imaging setup as for live imaging of COS-7 cells (above) was used.

**Quantification of the nanotube radius**. Nanotube radii were calculated from fluorescence intensity calibration of the lipid film on a flat surface[60]. Flat supported bilayer was used to find the density of the membrane fluorescence signal ($\rho_0$), then the nanotube radius was obtained from the total fluorescence per unit length of the nanotube $Fl$ using $r = Fl/2\pi\rho_0$.

**Quantification of the GFP-Rtnl1 sorting**. The sorting coefficient, defined as the relative change of the membrane area fraction occupied by GFP-Rtnl1 $\Delta\varphi/\varphi$ (where the area fraction is defined as $\varphi = \frac{a}{A+a}$, with $A + a$ is the membrane area per one Rtnl1 molecule and $a$ is the area occupied by the molecule, Supplementary Eq. (1)), was calculated using GFP-Rtnl1/Rh-DOPE fluorescence ratio[34]. We assumed that for a low-protein concentration the ratio of the protein to lipid fluorescence, per unit area, $F_{GFP-Rtnl1}/F_l = F^0_{GFP}/(A\rho_0) = \varphi F^0_{GFP}/(a\rho_0)$, where $F^0_{GFP}$ is the fluorescence of a single GFP and $\rho_0$ is the density of the lipid fluorescence, so that $\frac{\varphi^{GUV}}{\varphi^{nanotube}} = \frac{(F_{GFP-Rtnl1}/F_l)^{reservoir}}{(F_{GFP-Rtnl1}/F_l)^{nanotube}}$. $F_{GFP-Rtnl1}/F_l$ was measured as the ratio of total fluorescence intensities obtained from a ROI upon subtraction of the background, neglecting the polarization factor[34]. Two different circular ROIs (diameter 2 μm) were used for each reservoir/nanotube membrane.

**Force measurements with optical tweezers**. A counter propagating dual-beam optical tweezers instrument equipped with light-momentum force sensors was used in these experiments, which is capable of measuring force directly[61]. The two lasers are brought to the same focus through opposite microscope objective lenses generating a single optical trap. Protein–lipid nanotubes were generated in situ as follows. Pre-hydrated 5 μm polystyrene beads covered with proteolipid lamellas (as described above) were introduced into the experimental chamber containing 20 μL of working buffer at 22 ± 1 °C. One bead was hold in the optical trap and brought into contact with a 2 μm streptavidin-covered bead immobilized by suction in a micropipette tip. The two beads were separated with an initial constant pulling speed of 0.1 μm/s to form a tube. Extension-shortening cycles were performed on individual tubes at different pulling rates (as indicated in the main text). Below 8 μm s$^{-1}$ the trap was displaced linearly at a fixed calibrated speed. For higher velocities, the pipette was displaced by a coarse positioner away from the optical trap, and the pulling rates were calculated offline as the distance change per unit time. Data were collected with high force (<1 pN), position (1–10 nm) and temporal (500 Hz) resolutions. A similar procedure was performed with pure lipid films to test the behavior of protein-free tubes under force.

## Data availability

The data that support the findings of this study are available from the corresponding authors on reasonable request. The source data underlying Figs. 1d, 2b, 3e, 4b–d, 5b, d, f and Supplementary Figs. 1c–e, 2d, 4d, 5c, and 6b are provided as a Source Data file.

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

## Acknowledgements

This work was partially supported by NIH R01GM121725 to V.A.F., a 5×1000 grant from the Italian Ministry of Health and Telethon GGP11189 to A.D., Spanish Ministry of Science, Innovation and Universities grants BFU2015-70552-P to V.A.F. and A.V.S., and BFU2015-63714-R and PGC2018-099341-B-I00 to B.I., Basque Government grant IT1196-19, Russian Science Foundation Grant No. 17-75-30064 and Ministry of Science and Higher Education of the Russian Federation.

## Author contributions

Conceptualization: A.D., V.A.F., and A.V.S.; Methodology: A.D., V.A.F., A.V.S., D.P., and B.I.; Investigation: J.E., D.P., A.E., G.M., T.T., A.M., A.V.d.O., R.B., A.V.S.; theoretical analysis: V.A.F., P.I.K., P.V.B.; writing—original draft, A.D. and V.A.F.; writing—review & editing: A.D., V.A.F., J.E., A.V.S., and D.P.; funding acquisition: A.D., V.A.F., A.V.S., B.I., and P.V.B.; resources: A.D., V.A.F., B.I., and S.B.; supervision: A.D., V.A.F., A.V.S., and B.I.

## Competing interests

The authors declare no competing interests.

## Additional information

**Peer Review Information** *Nature Communications* thanks the anonymous reviewer/s for their contribution to the peer review of this work. Peer reviewer reports are available.

