## [Peer Review File · Nature Communications]

Reviewers' comments:

Reviewer #1 (Remarks to the Author):

The manuscript by Espadas, Pendin, Bocanegra et al. describe a phenomenon where lethality caused by the loss of ER-localized, fusion-promoting atlastin GTPases is rescued by the loss of ER-localized curvature-stabilizing reticulons. In combination with EM reconstructions of fly neurons and live cell microscopy of the larvae, rescue of lethality correlates with recovery of the characteristic network-like shape of the ER upon the combined loss of atlastin and reticulons, suggesting that these proteins function antagonistically to maintain ER form and function. The rescue of ER morphology in atlastin-deficient cells upon loss of reticulons is interpreted to emerge from a scenario whereby membrane fusion functions of atlastin counter the intrinsic tendency of reticulons-containing ER tubules to undergo fragmentation. Consistently, reticulon overexpression causes the ER to appear fragmented. In support of this model, the authors put together a set of challenging biophysical experiments that demonstrate that membrane nanotubes with reticulons exhibit fission when subjected to axial force upon elongation. Finally, a model suggesting that the presence of reticulons in the ER renders it susceptible to fragmentation in response to pulling forces exerted by molecular motor and that this represents an intrinsic mechanism to regulate organelle shape is put out to rationalize these results.

The findings in this paper are quite exciting and could potentially represent a novel mechanism for shape control of the ER. Data presented here and its analysis is of high quality and certainly deserves an audience. However, in the current form, the manuscript could tremendously benefit from control experiments and additional mechanistic insights that would validate the model proposed by the authors. Also, in general, a coherent discussion that stitches together the in vivo and in vitro results would improve the scope and quality of the manuscript.

Major comments:

The links made to ER form and dynamics in vivo to reticulon functions in vitro is a bold and exciting step. However, in its current form, the manuscript presents little corroborative evidence. The finding that lethality caused by the loss of atlastin is rescued by the loss of reticulon is indeed surprising. What is the cause for lethality in the first place? How does the ER appear in the absence of atlastins or is there any ER at all? Does the expression of a GTPase-dead mutant of atlastin cause rescue of lethality? Conversely, does the expression of mutant reticulons that are membrane inserted but deficient in stabilizing membrane curvature fail to rescue lethality. More importantly, do the same reticulon mutants when incorporated into model membranes also fail to display elongation-dependent fission of nanotubes. Studies of this kind would consolidate and establish the mechanism by which these proteins antagonise each other's functions and strengthen the proposed model.

With regard to the EM analysis, how is one certain that the organelle analyzed is the ER, especially in the absence of atlastins, since the changes to ER structure are expected to be quite dramatic. Was this analysis validated using ER markers?

Reticulon overexpression has earlier been reported to convert cisternal ER into long tubules (Voeltz, Cell, 2006), but that it causes ER fragmentation is a novel finding that deserves further analysis. How consistent and reproducible is this observation and does this reflect fragmented ER? Perhaps a figure panel of a group of cells with an ER marker with and without Rtnl-1-GFP could better represent this data rather than showing just one cell. Also, FLIP/EM experiments could confirm if the ER is indeed fragmented. Importantly, these results highlight a fundamental disconnect between native functions of reticulons and results obtained upon its overexpression. If reticulons were capable of such dramatic membrane remodeling, how then does one explain the existence of reticular ER in control cells, especially since reticulons are abundant ER-resident proteins? Is there a critical threshold concentration above which reticulons demonstrate a

tendency to fragment the ER. Can this be tested in vitro by titrating reticulon levels in nanotubes and testing at what concentration do they fail to exhibit elongation-induced fission? The time scales of fission appear to be very rapid (~ 10 s in the movie) while expression of Rtnl-1 should presumably take hours. It is therefore surprising (and possibly serendipitous) that ER fission is captured upon reticulon overexpression. Was this tried using an inducible expression system?

Better quantification of ER fragmentation is necessary to support the authors' claims that this is due to fission. Does the resultant ER show a loss of 3-way junctions or an absence of tubules altogether? Although arbitrary, can one correlate ER morphology to reticulon levels by plotting the number of 3-way junctions as a function of Rtnl-1-GFP fluorescence across multiple cells? How does one distinguish between growth and retraction of ER tubules, which is otherwise quite prevalent, from fission of a preformed tube? The current description of these results offers little detail to distinguish between these effects. Also, what is the retraction due to? Slower retraction could be due to friction but also due to global perturbation due to problems in engagement with motor proteins.

That reticulon-containing nanotubes undergo elongation-dependent fission is very interesting. However, this results is phenomenological with little mechanistic insight into how constriction and fission occurs. Is Rtnl uniformly present at both the bulged and constricted region on the nanotube? Does pulling lead to a change in protein distribution on the nanotube? It would be worthwhile to simultaneously plot changes in nanotube fluorescence (for constriction) and forces in the same graph to causally link the two effects.

The authors state "The extension-contraction cycle revealed a hysteresis in force-length dependence, large in Rtnl1- containing tubes and virtually absent in pure lipid nanotubes (Fig. 4d)." Is there any indication of protein redistribution in these experiments? Could such changes explain the slow dissipation of membrane tension in reticulon-containing nanotubes. What is the origin of the drag?

Also, the authors report that 15 out of 32 tubes underwent fission at $0.1 \mu\text{m/s}$ extension speed. Was there any difference in the pathway to membrane remodeling or reticulon distribution in the remaining 17 events? To call the fission process stochastic would require all 32 events to have behaved identically with respect to changes in membrane tension.

In addition, can the fission event be captured in finer detail, at least by fluorescence microscopy? Does the break in the nanotube occur at the bulge or the pre-constricted part of the tube?

Furthermore, the link to an elongation-dependent fission mechanism in cells is a tall claim since ER fission is never observed or reported under native conditions.

The methodology reported for reconstituting reticulons is novel and for this reason could do with more explanation and controls. Are the reticulons purified in detergent and then inserted into preformed LUVs? This was not clear from the methods section. The authors state that residual detergent is removed using adsorbent beads but Triton X-100 is notoriously difficult to remove. Could residual detergent in the membrane make them susceptible to pulling-induced rupture? Understandably, one cannot do without detergent but a control wherein membranes formed under similar conditions but without the addition of reticulons would substantially address this concern. Importantly, how do structural attributes of reticulons feed into this behavior? For instance, would the incorporation of a control protein with a single transmembrane domain at such concentrations cause a similar increase in force upon nanotube extension and undergo fission. The authors state, "The large membrane-inserting reticulon-homology domain, fully spanning the outer and partially the inner lipid leaflets, is likely to produce such a viscous drag, further enhanced by reticulon oligomerization. These points could be easily validated with such a control.

Minor comments:

Statements listed below could be modified to parse out interpretations made of actual data from speculation.

"Our findings imply that in a living cell, the ability of Rtnl1 to produce membrane constriction is inseparable...". There is little evidence that reticulons cause constriction of the ER in vivo.

"During elongation, the pulling force acted on such pre-constricted parts of the nanotube bringing their curvature close to the hemi-fission threshold (Fig. 4f, red) and thus enabling fission along an alternative, pore-free path". This is speculative since the current results do not rule out the possibility of membrane rupture.

In the abstract section, the authors state "Live imaging of ER network dynamics upon ectopic expression of drosophila reticulons linked fission to augmented membrane friction". What do the authors mean by this statement? If this refers to slower retraction of the cut ends of the ER seen upon reticulon overexpression then this is need to be modified based on previous comments.

Also, the sentence "...novel mechanism of organelle regulation where the dynamic balance between fission and fusion is governed by motility" is perhaps incorrect. It is the balance between organelle fission and stretch-dependent fission that governs organelle morphology.

On pg. 13, "Interaction between the two constriction modes is more complex near the reservoir....". Please clarify as to which of the two modes is the reference made to.

Pg. 3, second sentence from bottom, construction should be constriction.

How are the plots in Fig. 4D and E different? Could they be combined into one figure?

There's no movie 10.

Pg. 9, Nanotubes mimic the ER tubes and not branches

In Figure 2D and E, please label in figure and legend what the top and bottom panels are?

Reviewer #2 (Remarks to the Author):

Espadas et al propose the concept of the connected ER tubule network as a balance between fission by reticulons and fusion by atlastins. They show that imbalances in this relationship in drosophila result in ER fragmentation. In vitro experiments further demonstrate that reticulon proteins combined with physiological pulling forces can result in tubule fission.

The surprising result showing that a reticulon null mutation greatly rescues an atlastin null mutation is a compelling basis for the authors' hypothesis. The data convincingly supports reticulon and atlastin as antagonistic mechanisms. However, the in vitro experiments testing the role of reticulons as fission are not clear.

Major concerns:

1. The in vitro experiments shown in figure 4 could test either tubule fission (as intended) or tubule extension. Both fission and extension events include reticulon proteins and include forces pulling on membranes. If the types of forces exerted on the tubules is different, please clearly describe the differences so one can distinguish between them as many of the in vitro tubule experiments seem to also test for the durability of extending tubules.

Minor concerns:

1. Figure 2A shows a STED image of a large field of view ($>10\ \mu\text{m}$). Since STED is a super resolution method, one expects a more zoomed in image showing a 50-250 nm-sized feature. Furthermore, STED imaging cannot improve your resolution unless the appropriate pixel size is used. Please include the pixel size in your methods section.
2. Figure 2A shows bright domains as indicators of ER network imbalance, but there is no discussion as to the morphology of these domains, other than as presumed disconnected ER elements. Could these bright domains be ER tubules closely positioned near the nucleus, residing where sheets usually are? Discussion of axial resolution and relative brightness could be used to inform the morphology of these structures.
3. Figure 3A "Rtnl1 OE" shows a middle section of a cell, where the ER does not appear to be well connected. This apparently disconnected ER may result from taking a middle section of the cell. Replace this image with a region closer to the coverglass to be a better comparison to the control cell.
4. Previously published in vitro experiments show the concentration of curvature stabilizing proteins dictates the degree of curvature in ER tubules. Are tubules breaking at the region with the thinnest diameter? If yes, do fission points also have the highest concentration of reticulon?
5. The authors report that "Interaction between the two constriction modes is more complex near the reservoir membrane, where elongating nanotubes appeared visibly thinner (Fig 4c)." In vitro tubule diameters reported as nanotube radius (extended data 3C) do not indicate any variation along tubule length. It appears that the authors have tools address the relation between tubule diameter and fission point, or even tubule diameter in extending vs. fission-event tubules. Further, the authors can bring this into context of physiological ER tubule diameters.

RESPONSE TO THE EDITOR AND REVIEWERS

We would like to thank all reviewers for taking the time to examine and evaluate our manuscript and for the key suggestions they have provided to make the manuscript stronger and more convincing.

Editor: Notably, we would require better clarification of ER morphology in atlastin loss-of-function (please see comments from Reviewer #1).

We would like to point out that the atlastin null mutant was extensively characterized earlier (Orso et al, Nature, 2009). In this study we used exactly the same scoring method. We also emphasize that the major result here is not the atlastin loss of function phenotype but the reappearance of a wt phenotype upon depletion of Rtnl1 in the *atl*² mutant background. In *Rtnl1*¹/*atl*² double mutants, ER profile length (a measure of Atl/Rtnl balance) is indistinguishable from the control. And this result was accepted without questions by both reviewers.

Regarding atlastin null mutants, we wish to underscore that changes are indeed dramatic but not so much so that the ER is no longer recognizable when present. Nevertheless, to address directly reviewer's #1 concern about recognition of the ER, we add to our point-by-point response an additional verification of the correct ER detection in wt and *atl*² neurons based on the use of a *Drosophila* version of ER FLIPPER¹, an HRP-based luminal marker that labels the ER with a black precipitate.

Editor: Furthermore, we would require the effects of reticulon overexpression on ER fragmentation to be further clarified with better quantification, distinguishing between growth and retraction, and comparison of reticulon levels (reviewer #1).

First, we would like to note that the changes of ER morphology and topology upon Rtnl1 overexpression in *Drosophila*, our main *in vivo* model, have been quantified using EM, STED microscopy and FLIP. STED analysis demonstrates ER constriction (seen as redistribution of the ER into bright punctae, Fig. 2a-c) and FLIP analysis further linked the constriction to ER fragmentation (Fig. 2d). Importantly, the outcomes of overexpression were analyzed at different levels of atlastin, revealing the antagonistic effects of Rtnl1 and atlastin on ER morphology and connectedness. In the revised version of the manuscript we provide additional quantification of this antagonistic interaction (Supplementary Fig. 1d, e) and also include a simple kinetic model describing the antagonism (Supplementary Information).

In this context, we consider the COS-7 overexpression as an auxiliary model suitable for visualization of dynamic transition from the ER constriction to fragmentation but not for analyses of the end-state fragmentation. The time progression of the ER constriction driven by Rtnl1 in COS-7 model is now quantified using time-dependent redistribution of an ER luminal marker between ER tubules and bright fluorescent domains (Supplementary Fig. 2 a, b), similarly to STED analysis in *Drosophila* (Fig. 2a-c). We found that such ratiometric quantification reveals different ER-remodeling power of Rtnl1 and GFP-Rtnl1 (Supplementary Fig. 2 c,d). We used a plasmid tandemly expressing Rtnl1 and nuclear CFP so that the Rtnl1 levels could be estimated from CFP fluorescence (Supplementary Fig. 2a) and a plasmid expressing only GFP-Rtnl1 whose expression levels were estimated from GFP fluorescence. Though we could not directly compare the Rtnl1 and GFP-Rtnl1 levels, we found a threshold-type time dependence of ER constriction by these proteins (Supplementary Fig. 2d), leading to drastically different levels of ER fragmentation by Rtnl1 (complete fragmentation 24h PT, Supplementary Fig. 2b) and GFP-Rtnl1 (ER network remains visible 24h PT, Supplementary Fig. 2b). Importantly, *in vitro* analysis of Rtnl1 and GFP-Rtnl1 revealed similar threshold-type dependence of the fission activity of the proteins on their concentration, with the threshold shifted to higher concentrations for GFP-Rtnl1 (Supplementary Fig. 6a), consistent with impairment of the curvature-creation activity of the protein (Supplementary Fig. 6b). We believe that this novel link between *in vitro* and cellular datasets strongly substantiates our approach and model.

Finally, we now explain in detail our criteria for fission detection and how we discriminate that from growth-retraction cycles of dynamic ER tubules (Fig. 3d).

Editor: Both Reviewers #1 and #2 share the concern that the role of tubule fission as opposed to extension is currently unclear.

Our message, explained in more details in the new version of the manuscript, is that these two roles are not opposed. Rather curvature creation and fission of tubular ER branches by reticulon are inseparable aspects of ER membrane dynamics. We note that ER “tubulation” (conversion of the ER sheets into tubules) in reticulon-expressing cells has been linked to ER fragmentation since the article by the Rapoport group (Voeltz et al. Cell 2006, a quote from this article “Expression of hemagglutinin (HA) tagged Rtn1p under an inducible Gal promoter resulted in disruption of the peripheral ER”). Multiple reports on reticulon-driven fragmentation followed (cited in the manuscript), including the recent work of the Rapoport group pointing directly to reticulon as the probable driver of the ER membrane scission (Wang et al., Elife, 2016). Our analyses provide a first (to the best of our knowledge) mechanistic explanation as to how a single protein can do both, creation and disruption of tubular membrane geometry. While the creation is based on the well-known membrane wedging phenomenon, disruption is explained by a novel mechanism, constriction-by-friction. This disruption relies on membrane dynamics so that in still membranes reticulons function as curvature creators/stabilizers while in dynamic ER branches reticulon can stochastically mediate fission. To further clarify the issue, we have elaborated a simple kinetic model explaining how the two activities of Rtn11 converge with the fusogenic activity of atlastin in ER maintenance. This new model is now included in the manuscript.

Reviewer #1(Remarks to the Author):

The manuscript by Espadas, Pendin, Bocanegra et al. describe a phenomenon where lethality caused by the loss of ER-localized, fusion-promoting atlastin GTPases is rescued by the loss of ER-localized curvature-stabilizing reticulons. In combination with EM reconstructions of fly neurons and live cell microscopy of the larvae, rescue of lethality correlates with recovery of the characteristic network-like shape of the ER upon the combined loss of atlastin and reticulons, suggesting that these proteins function antagonistically to maintain ER form and function. The rescue of ER morphology in atlastin-deficient cells upon loss of reticulons is interpreted to emerge from a scenario whereby membrane fusion functions of atlastin counter the intrinsic tendency of reticulons-containing ER tubules to undergo fragmentation. Consistently, reticulon overexpression causes the ER to appear fragmented. In support of this model, the authors put together a set of challenging biophysical experiments that demonstrate that membrane nanotubes with reticulons exhibit fission when subjected to axial force upon elongation. Finally, a model suggesting that the presence of reticulons in the ER renders it susceptible to fragmentation in response to pulling forces exerted by molecular motor and that this represents an intrinsic mechanism to regulate organelle shape is put out to rationalize these results.

The findings in this paper are quite exciting and could potentially represent a novel mechanism for shape control of the ER. Data presented here and its analysis is of high quality and certainly deserves an audience. However, in the current form, the manuscript could tremendously benefit from control experiments and additional mechanistic insights that would validate the model proposed by the authors. Also, in general, a coherent discussion that stitches together the in vivo and in vitro results would improve the scope and quality of the manuscript.

We thank the reviewer for positive evaluation of our work and the many constructive comments. We performed additional experimental and theoretical analyses to address her/his critique. We trust that

the new mechanistic insights, control experiments and expanded discussion presented in the revised version of the manuscript will be satisfactory.

Reviewer #1, Major comments:

The links made to ER form and dynamics in vivo to reticulon functions in vitro is a bold and exciting step. However, in its current form, the manuscript presents little corroborative evidence. The finding that lethality caused by the loss of atlastin is rescued by the loss of reticulon is indeed surprising. What is the cause for lethality in the first place? How does the ER appear in the absence of atlastins or is there any ER at all? Does the expression of a GTPase-dead mutant of atlastin cause rescue of lethality?

We note that the changes of ER morphology caused by loss of *at1* and by expression of a GTPase-dead *at1* mutant have been thoroughly characterized earlier (Orso et al, Nature, 2009). Here we do not focus on the *at1* null phenotype per se. Rather, we show how changes in Rtnl1 levels affect ER morphology in the *at1* null background. Strikingly, we found that elimination of atlastin reveals the intrinsic constriction and fission activities of Rtnl1 in the ER, already at physiological levels of the protein.

As for the lethality due to loss of atlastin, it is a genetic observation demonstrated earlier (Orso et al, Nature, 2009). The mechanisms of the lethality are yet to be determined, the required analyses of the organism dysfunction(s) are out of the scope of this work. Here we use the lethality only as a reporter of genetic interactions between *at1* and Rtnl1.

Conversely, does the expression of mutant reticulons that are membrane inserted but deficient in stabilizing membrane curvature fail to rescue lethality.

We note that the depletion, not (over)expression of Rtnl1 in *Atl²* null background (*Rtnl1¹/Atl²* double mutant) rescued the *Atl²* lethality (Fig. 1a). Removal of Rtnl1 alone has no effect on the adult survival (see Fig 1a). Hence the situation is complex and related to dynamic balance between the Rtnl1 and *At1* activities, summarized by a simple kinetic model in the new version of the manuscript (see Supplementary Information).

More importantly, do the same reticulon mutants when incorporated into model membranes also fail to display elongation-dependent fission of nanotubes. Studies of this kind would consolidate and establish the mechanism by which these proteins antagonise each other's functions and strengthen the proposed model.

In the revised version of the manuscript we address this important concern. We noticed earlier that GFP-Rtnl1 ability to constrict and disrupt the ER is inferior to that of the wild type protein. We analyzed the effect quantitatively and revealed impairment of the ER constriction/fragmentation activities of GFP-Rtnl1 at early stages post-transfection (up to 24h PT, Supplementary Fig. 2b-d). We next found an analogous reduction of the fission activity of GFP-Rtnl1 *in vitro* (Supplementary Fig. 6). Notably, the dependency of fission/fragmentation activities on protein concentration display different thresholds for Rtnl1 and GFP-Rtnl1 (Supplementary Fig. 2d and 6a), with larger amount of GFP-Rtnl1 needed to produce membrane transformation both *in vivo* and *in vitro*. We believe that such strong correlation between *in vitro* and *in vivo* datasets validates our approach and strengthens our model. We also note that modification of the N-terminus of a member of reticulon family, Yop1p, was previously utilized to link *in vitro* and *in vivo* activities of the protein².

We would like to stress here that controlled impairment of Rtnl1 fission activity by rational mutagenesis of Rtnl1 would be a complex and lengthy project. The reason being that, to the best of our knowledge, there is no published strategy for reticulon mutagenesis specifically aiming at quantitative alteration of its curvature activity. The RHD domain, primarily implicated in membrane curvature production, was also associated with reticulon oligomerization and thus, as our results indicate, is expected to affect visco-elastic properties of the membrane. Though some alterations of

RHD (e.g. changing its length) have been tested in overexpression models, they yielded little mechanistic insights. Even RHD topology in the membrane remains a subject of debate. Hence, while we intend to develop our research in this direction, the theme clearly constitutes a separate long-term project.

With regard to the EM analysis, how is one certain that the organelle analyzed is the ER, especially in the absence of atlastins, since the changes to ER structure are expected to be quite dramatic. Was this analysis validated using ER markers?

[redacted]

[redacted]

[redacted]

Reticulon overexpression has earlier been reported to convert cisternal ER into long tubules (Voeltz, Cell, 2006), but that it causes ER fragmentation is a novel finding that deserves further analysis. How consistent and reproducible is this observation and does this reflect fragmented ER? Perhaps a figure panel of a group of cells with an ER marker with and without Rtnl-1-GFP could better represent this data rather than showing just one cell. Also, FLIP/EM experiments could confirm if the ER is indeed fragmented. Importantly, these results highlight a fundamental disconnect between native functions of reticulons and results obtained upon its overexpression.

We note that even Voeltz et al (Cell 2006) say that “Expression of hemagglutinin (HA) tagged Rtn1p under an inducible Gal promoter resulted in disruption of the peripheral ER”. Hence, the observation that reticulon overexpression could go all the way to ER disruption has been reported by different labs (as cited in our manuscript).

Here we analyze the effect of Rtnl1 overexpression on the ER morphology/topology in two different models. Our main model is *Drosophila*, where we detected and analyzed the ER fragmentation using a combination of techniques: EM, STED fluorescence microscopy and FLIP. In this system we could link the ER fragmentation observed upon loss of atlastin to the unopposed endogenous activity of Rtnl1. These results demonstrate, for the first time, that the ER fragmentation activity is intrinsic to Rtnl1 (not an over-expression artifact). In a stationary ER network, the fragmentation activity is balanced by the fusogenic activity of atl to maintain ER morphology (see Fig.1b and also the kinetic model in Supplementary Information, p.9)

The second model, Rtnl1 overexpression in COS-7 cells, is auxiliary. The COS-7 model was primarily intended for live imaging of Rtnl1-containing ER network with the main purpose to detect and analyze individual events of ER fragmentation, such as scission of the ER branches, impossible to resolve in living fly tissues. However, as requested, now we present groups of representative images showing the ER disruption in COS-7 by Rtnl1, 24h after transfection (Supplementary Fig. 2b). Furthermore, we also quantified the time progression of the ER constriction leading to the ER

disruption. For that we used a generic metric based upon changes of fluorescence intensity distribution (per pixel) of a luminal ER marker during ER constriction (Supplementary Fig. 2c,d).

If reticulons were capable of such dramatic membrane remodeling, how then does one explain the existence of reticular ER in control cells, especially since reticulons are abundant ER-resident proteins? Is there a critical threshold concentration above which reticulons demonstrate a tendency to fragment the ER. Can this be tested *in vitro* by titrating reticulon levels in nanotubes and testing at what concentration do they fail to exhibit elongation-induced fission?

Yes, our results indicate that certain minimal/threshold concentration of Rtnl1 in the membrane is required to trigger ER fragmentation *in vivo* as well as tubular membrane scission *in vitro* (Supplementary Fig. 2d, 6a). Our results further demonstrate that at 1:150 Rtnl1/lipid ratio Rtnl1 creates static membrane curvatures comparable to that observed in the tubular ER (according to published data, e.g. see Hu et al., Science 2008 cited in the manuscript) and robustly produced membrane fission. We believe that this protein/lipid ratio represents physiological combination of curvature stabilization/fission activities by Rtnl1. Why the ER network remains reticular then? First, we note that the ER network fragments upon suppression of the fusogenic activity of At1 (Fig. 1b, Fig. 2), a phenomenon also observed in *in vitro* ER network mimics by the Rapoport group (Powers et al., Nature, 2017). Apparently, the fusogenic activity of at1 is required to maintain a reticular ER. Our manuscript highlights the utmost physiological importance of this balance not only for preservation of the ER morphology but also for the wellbeing of the whole organism (see Fig. 1a, b). Second, in the revised manuscript we emphasize the stochastic nature of the fission process and transient character of curvature stresses leading to fission (Supplementary Fig. 5d). These results demonstrate that the fission probability is controlled not only by Rtnl1 concentration but also by a set of parameters related to ER dynamics (such as characteristic rates and lengths of the translocation of ER elements), enabling versatile physiological regulation of the ER fragmentation.

The time scales of fission appear to be very rapid (~10 s in the movie) while expression of Rtnl-1 should presumably take hours. It is therefore surprising (and possibly serendipitous) that ER fission is captured upon reticulon overexpression. Was this tried using an inducible expression system?

We found that the ER fragmentation surged at 12-17h post-transfection. We looked for cells showing ER constriction in progress (we scored for the emergence of the blobs, as in Supplementary Fig. 2c,d, and the amount of Rtnl1 expressed, as in Supplementary Fig. 2a). This approach allowed us to detect membrane scission events in the ER. We note that scission of ER tubules has been recently reported by using cutting-edge SR imaging methods (Guo et al, 2018). Hence, our method is likely to underestimate the number of fission events.

Better quantification of ER fragmentation is necessary to support the authors' claims that this is due to fission. Does the resultant ER show a loss of 3-way junctions or an absence of tubules altogether? Although arbitrary, can one correlate ER morphology to reticulon levels by plotting the number of 3-way junctions as a function of Rtnl-1-GFP fluorescence across multiple cells?

We politely disagree. Our main assay for fragmentation is FLIP. The luminal marker we used, KDEL-GFP, has a gyration radius of a GFP molecule, ~2.5nm. Constriction of a membrane tube to 4nm (luminal diameter) causes spontaneous fission (e.g. see works by M. Kozlov and earlier work of the authors, Shnyrova et al, Science, 2013), hence complete blockage of KDEL-GFP migration through the ER network strongly indicates membrane scission. Unable to perform live imaging of fly tissues to directly confirm fission, we opted for the COS-7 model where both fragmentation and fission were detected. As for ER morphology (tubulation), we use ER profile length as a universal metrics for At1-

Rtnl1 antagonism. Although we agree that counting 3-way junctions should help characterizing the ER morphology, we are not sure how such quantification could advance our particular study.

How does one distinguish between growth and retraction of ER tubules, which is otherwise quite prevalent, from fission of a preformed tube? The current description of these results offers little detail to distinguish between these effects. Also, what is the retraction due to?

In the revised version of the manuscript we compare fission and growth-retraction events and explain how we discriminate one from the other (Fig. 3d). We believe that retraction is driven by the intrinsic tension in the ER membrane system, e.g. due to interaction with microtubule network and motor pulling action.

Slower retraction could be due to friction but also due to global perturbation due to problems in engagement with motor proteins.

We agree and changed the text accordingly.

That reticulon-containing nanotubes undergo elongation-dependent fission is very interesting. However, this results is phenomenological with little mechanistic insight into how constriction and fission occurs

In fact, we performed extensive theoretical analysis of the phenomenon, which we call constriction-by-friction. The theoretical treatment of curvature driven sorting of Rtnl1 in static and dynamic tubes, as well as the associated shear-thinning effect, are described in the Supplementary Information section of the revised manuscript.

Is Rtnl uniformly present at both the bulged and constricted region on the nanotube? Does pulling lead to a change in protein distribution on the nanotube? It would be worthwhile to simultaneously plot changes in nanotube fluorescence (for constriction) and forces in the same graph to causally link the two effects.

We followed the advices and analyzed the curvature-driven redistribution of Rtnl1 and GFP-Rtnl1 both experimentally (using different readouts from both fluorescence microscopy and force relaxation measurements, see new Fig. 4 and 5) and theoretically. We made the correlative curvature/force plot to better illustrate the link between the nanotube constriction and the force increase (Fig. 5a).

The authors state "The extension-contraction cycle revealed a hysteresis in force-length dependence, large in Rtnl1- containing tubes and virtually absent in pure lipid nanotubes (Fig. 4d)." Is there any indication of protein redistribution in these experiments? Could such changes explain the slow dissipation of membrane tension in reticulon-containing nanotubes. What is the origin of the drag?

The revised version of the manuscript now includes the analysis of steady-state redistribution of GFP-Rtnl1 between the reservoir and the nanotube membranes (new Fig. 4c). As in the previous version, we associated the slow dissipation of membrane tension (Supplementary Fig. 5b) with Rtnl1 sorting towards the nanotube. Importantly, the sorting coefficients obtained from fluorescent microscopy measurements for GFP-Rtnl1 was substantially smaller than that obtained from the force measurements for non-labeled Rtnl1. Consistently, GFP-Rtnl1 is impaired in both membrane curvature creation and fission (both in vitro and in over-expression model, Supplementary Fig. 2b-d, 5c, 6b). Accordingly, we could not resolve the changes of GFP-Rtnl1 redistribution during extension-contraction cycles as the curvature changes and associated changes in GFP-Rtnl1 concentration were too small to be detected by fluorescence microscopy. Finally, we respectfully note that also the original theoretical analysis associated the viscous drag with increased viscosity of Rtnl1-containing membranes.

Also, the authors report that 15 out of 32 tubes underwent fission at 0.1 $\mu\text{m/s}$ extension speed. Was there any difference in the pathway to membrane remodeling or reticulon distribution in the remaining 17 events? To call the fission process stochastic would require all 32 events to have behaved identically with respect to changes in membrane tension.

We agree. To substantiate the conjecture of the stochastic nature of scission we now explicitly compare the stresses in broken and unbroken tubes (on p.12). We also explored the transient character of stress during limited-length elongation and revealed that incremental elongation can proceed without substantial increment of the stress (Supplementary Fig. 5d).

In addition, can the fission event be captured in finer detail, at least by fluorescence microscopy? Does the break in the nanotube occur at the bulge or the pre-constricted part of the tube?

We performed the requested analysis. We found that fission always happens near membrane reservoirs at the ends of the nanotube and that the nanotube regions near the reservoir are more curved than those far from the reservoir (Fig. 5b).

Furthermore, the link to an elongation-dependent fission mechanism in cells is a tall claim since ER since fission is never observed or reported under native conditions.

We wish here to draw attention to the recent study by Guo et al, (Cell, 2018) reporting ER fission events detected by super-resolution fluorescence microscopy.

The methodology reported for reconstituting reticulons is novel and for this reason could do with more explanation and controls. Are the reticulons purified in detergent and then inserted into preformed LUVs? This was not clear from the methods section. The authors state that residual detergent is removed using adsorbent beads but Triton X-100 is notoriously difficult to remove. Could residual detergent in the membrane make them susceptible to pulling-induced rupture?

Understandably, one cannot do without detergent but a control wherein membranes formed under similar conditions but without the addition of reticulons would substantially address this concern.

We now explain in the Methods section that we made control, pure lipid nanotubes the same way we made proteo-lipid nanotubes. We used LUVs as the starting material. We passed the vesicles through the mock protein reconstitution procedure where we adjusted the detergent concentration to the same level as with Rtnl1 reconstitution. We also note the large difference in fission efficiency between purified Rtnl1 and GFP-Rtnl1, both proteins reconstituted along the same protocol.

Importantly, how do structural attributes of reticulons feed into this behavior? For instance, would the incorporation of a control protein with a single transmembrane domain at such concentrations cause a similar increase in force upon nanotube extension and undergo fission. The authors state, "The large membrane-inserting reticulon-homology domain, fully spanning the outer and partially the inner lipid leaflets, is likely to produce such a viscous drag, further enhanced by reticulon oligomerization. These points could be easily validated with such a control.

As mentioned above, such control experiments would not be straightforward. RHD constitutes the major part of the protein and thus has been implicated in curvature creation, oligomerization and specific interactions with lipids and cholesterol. Nevertheless, we found that Rtnl1 oligomerization might be sensitive to modification of its N-terminus, also reported earlier for Yop1p, another membrane of reticulon family (Hu et al., Science 2008). The revised manuscript contains initial mechanistic comparison of Rtnl1 and GFP-Rtnl1. Although we are keen to expand our research in this direction, it would not be feasible to accomplish such a task within the timeframe of this manuscript.

Reviewer #1, minor comments:

Statements listed below could be modified to parse out interpretations made of actual data from speculation.

"Our findings imply that in a living cell, the ability of Rtn1 to produce membrane constriction is inseparable...". There is little evidence that reticulons cause constriction of the ER in vivo.

We politely disagree. The reports are many, we provide a few quotes from leading groups in the field: "We also show that a complete reticulon homology domain is required for both RTN residence in high-curvature ER membranes and ER tubule constriction," *Plant Cell*. 2010 Apr; 22(4): 1333–1343. (Hawes group)

"Short hairpin TM domains of Rtn4a are required to constrict ER tubules" *Traffic*. 2011 Jan; 12(1): 28–41. (Voeltz group)

"whereas endogenous Rtn4 colocalized with the luminal protein calreticulin in mammalian cells, overexpressed Rtn4a, DP1, Rtn3c, or Yop1p squeezed calreticulin out of the tubules (Fig. 3C and fig. S7). *Science*. 2008 Feb 29;319(5867):1247-50. (Rapoport group)

"During elongation, the pulling force acted on such pre-constricted parts of the nanotube bringing their curvature close to the hemi-fission threshold (Fig. 4f, red) and thus enabling fission along an alternative, pore-free path". This is speculative since the current results do not rule out the possibility of membrane rupture.

In the revised version we provide additional experimental support for the hemi-fission (Fig. 5f).

In the abstract section, the authors state "Live imaging of ER network dynamics upon ectopic expression of drosophila reticulons linked fission to augmented membrane friction". What do the authors mean by this statement? If this refers to slower retraction of the cut ends of the ER seen upon reticulon overexpression then this is need to be modified based on previous comments.

We agree and corrected the sentence.

Also, the sentence "...novel mechanism of organelle regulation where the dynamic balance between fission and fusion is governed by motility" is perhaps incorrect. It is the balance between organelle fission and stretch-dependent fission that governs organelle morphology.

We agree and corrected the sentence.

On pg. 13, "Interaction between the two constriction modes is more complex near the reservoir....". Please clarify as to which of the two modes is the reference made to.

We changed the sentence to clarify the issue (p.11, revised manuscript)

Pg. 3, second sentence from bottom, construction should be constriction.

Fixed

How are the plots in Fig. 4D and E different? Could they be combined into one figure?

We opted to move 4D to the supplementary section as it indeed breaks the narrative.

There's no movie 10

Fixed

Pg. 9, Nanotubes mimic the ER tubes and not branches

Fixed

In Figure 2D and E, please label in figure and legend what the top and bottom panels are?
Fixed

Reviewer #2 (Remarks to the Author):

Espadas et al propose the concept of the connected ER tubule network as a balance between fission by reticulons and fusion by atlastins. They show that imbalances in this relationship in *Drosophila* result in ER fragmentation. In vitro experiments further demonstrate that reticulon proteins combined with physiological pulling forces can result in tubule fission. The surprising result showing that a reticulon null mutation greatly rescues an atlastin null mutation is a compelling basis for the authors' hypothesis. The data convincingly supports reticulon and atlastin as antagonistic mechanisms. However, the in vitro experiments testing the role of reticulons as fission are not clear.

We thank the reviewer for the overall positive opinion on our study.

Major concerns:

1. The in vitro experiments shown in figure 4 could test either tubule fission (as intended) or tubule extension. Both fission and extension events include reticulon proteins and include forces pulling on membranes. If the types of forces exerted on the tubules is different, please clearly describe the differences so one can distinguish between them as many of the in vitro tubule experiments seem to also test for the durability of extending tubules.

This is an excellent point. In the revised version of the manuscript we explicitly compare the forces exerted on broken and unbroken tubes under similar pulling conditions. The maximal force values are statistically undistinguishable, pointing to the stochastic nature of membrane fission under curvature stress. We note that the probabilistic nature of scission is due to limited amount of stress applied to the nanotubes during pulling with physiologically relevant speeds and at the physiologically-relevant amounts of Rtn11, as explained in the Concluding Remarks section of the revised manuscript. At extremely high speeds (100 μ m/s) all of the proteo-nanotubes broke despite the short time under stress.

Minor concerns: 1. Figure 2A shows a STED image of a large field of view (>10 μ m). Since STED is a super resolution method, one expects a more zoomed in image showing a 50-250 nm-sized feature. Furthermore, STED imaging cannot improve your resolution unless the appropriate pixel size is used. Please include the pixel size in your methods section.

We included the pixel size information. We could not resolve the fine structure of the ER in *Drosophila* neuron cells (specifically, the morphology of the bright puncta) so we opted not to include zoomed images as they carry no additional information.

2. Figure 2A shows bright domains as indicators of ER network imbalance, but there is no discussion as to the morphology of these domains, other than as presumed disconnected ER elements. Could these bright domains be ER tubules closely positioned near the nucleus, residing where sheets usually are? Discussion of axial resolution and relative brightness could be used to inform the morphology of these structures.

At this stage, we can only speculate about the morphology of the domains. Our preliminary EM analysis revealed highly convoluted ER subnetworks emerging upon Rtn11 over-expression but we are yet to explicitly relate those to the bright fluorescent domains. We note that the origin and structure of these domains are not directly related to the scope of our manuscript.

3. Figure 3A "Rtn11 OE" shows a middle section of a cell, where the ER does not appear to be well connected. This apparently disconnected ER may result from taking a middle section of

the cell. Replace this image with a region closer to the coverglass to be a better comparison to the control cell.

We apologize for the confusion. In fact, we used epi-fluorescence microscopy in live imaging of COS-7 cells, primarily, to resolve individual fission events (as now correctly described in Methods section). We used still images from the image stacks to quantify ER constriction and fragmentation, as shown now in the new Supplementary Fig. 2. There, we provide several representative images illustrating complete disruption of the ER in Rtnl1-expressing COS-7 cells at 24h PT.

4. Previously published in vitro experiments show the concentration of curvature stabilizing proteins dictates the degree of curvature in ER tubules. Are tubules breaking at the region with the thinnest diameter? If yes, do fission points also have the highest concentration of reticulon?

We note that the “static” curvature created by Rtnl1 (without assistance from pulling forces) is not sufficient to trigger fission, as explained now on p.9 of the revised manuscript. In vitro, elongation (pulling) of Rtnl1-containing tubes leads to their constriction, higher near membrane reservoirs where fission always occurs (Fig. 5b). In vivo fission also follows local pre-constriction (Fig. 3c,d). However, this pre-constriction is less than 2-fold (Fig. 5a, b) and hence is not causing detectable redistribution of labeled Rtnl1.

5. The authors report that “Interaction between the two constriction modes is more complex near the reservoir membrane, where elongating nanotubes appeared visibly thinner (Fig 4c).” In vitro tubule diameters reported as nanotube radius (extended data 3C) do not indicate any variation along tubule length. It appears that the authors have tools address the relation between tubule diameter and fission point, or even tubule diameter in extending vs. fission-event tubules. Further, the authors can bring this into context of physiological ER tubule diameters.

We now support the claim with quantification of the constriction enhancement near the reservoirs (Fig. 5b). We also relate the static curvature created by Rtnl1 to that of the ER tubules and conclude that at 1:150 Rtnl1/lipid ratio Rtnl1 produces both physiologically relevant static membrane curvature and fission (see Concluding Remarks).

References

- 1 Kuipers, J. *et al.* FLIPPER, a combinatorial probe for correlated live imaging and electron microscopy, allows identification and quantitative analysis of various cells and organelles. *Cell Tissue Res* **360**, 61-70, doi:10.1007/s00441-015-2142-7 (2015).
- 2 Hu, J. *et al.* Membrane proteins of the endoplasmic reticulum induce high-curvature tubules. *Science* **319**, 1247-1250 (2008).

Reviewers' comments:

Reviewer #1 (Remarks to the Author):

The revised manuscript address all of my initial concerns and puts forth new data that corroborates and reinforces the authors' claims. I have no further comments and recommend publication.

Reviewer #2 (Remarks to the Author):

The authors present data supporting the remarkable genetic interaction where a Rtnl1 knockout rescues the lethality phenotype of an at1 knockout. In vitro experiments suggest that the basis of this genetic interaction is due to the role of Rtnl1 in ER tubule fission. The basis of the manuscript is clearly defined and the results are compelling.

Revisions to their original manuscript enhance the original findings, showing the ER fragmentation by Rtnl1 in COS7 cells is more severe the longer Rtnl1 is over expressed. The discussion of GFP-Rtnl1 vs untagged Rtnl1 fission activity is useful for interpreting in vivo and in vitro results. And discussion of tubule diameter, Rtnl1 concentration, and fission as well as discussion of static versus force dynamics on tubules enhances the interpretation of the in vitro system.

ER tubule fission is not well documented in the existing ER dynamics literature and, from my perspective, not generally accepted as a primary mechanism for ER network maintenance (usually overshadowed by tubule fusion, tubule sliding, and ring closure). The authors acknowledge this in the current manuscript, "The stochastic character of fission likely explains the low occurrence of fission events in the ER under normal circumstances when Rtnl1 concentration and pulling forces and speeds are within physiological limits." However, tubule fission was recently shown by Guo et al 2018 Cell (mentioned twice by the authors in their rebuttal) and should be acknowledged in this manuscript. The findings by Espadas et al are notable and important for understanding of ER dynamics, morphology, and related disease. In response to their revisions and rebuttal, I suggest the following requirements before publication.

Requirements for publication:

1. ER tubule fission events are rare and under appreciated. ER tubule fission is not well known/accepted in the field of ER morphology. Please cite Guo et al 2018 Cell in the introduction. This would fit near the discussion of fragmentation and tubule fission. This specific paper is cited by Espadas et al in the revision rebuttal twice, since it contains examples of live mammalian ER tubule networks imaged at high speeds with specific instances of tubule fission. Citing another peer reviewed publication that used a different fluorescence imaging approach normalizes under appreciated ER fission events and elevates the current manuscript which offers a mechanism for these fission events.
2. Please add some discussion of in vitro tubule diameter shown in Figure 4b and how it relates to physiological ER tubule diameters (physiological diameter measured by Shim et al 2012 PNAS, Terasaki 2018 J Cell Sci, Schroeder et al 2019 J Cell Bio). Interestingly, neuronal ER tubules are reported to have characteristically narrow ER tubules, which likely is relevant to rates of fission.
3. I estimate their STED resolution is close to 150 nm, which is more in the domain of structured illumination (SIM) than standard STED implementation (40-80 nm resolution). I would recommend to them that they should use confocal imaging without heavy deconvolution for Fig2a and b, which I expect will produce results similar to that found using their STED images.

RESPONSE TO REVIEWERS

Reviewer #1 (Remarks to the Author):

We are grateful to the reviewer for appreciating our revisions and recommending publication

Reviewer #2 (Remarks to the Author):

The authors present data supporting the remarkable genetic interaction where a Rtnl1 knockout rescues the lethality phenotype of an atl knockout. In vitro experiments suggest that the basis of this genetic interaction is due to the role of Rtnl1 in ER tubule fission. The basis of the manuscript is clearly defined and the results are compelling. Revisions to their original manuscript enhance the original findings, showing the ER fragmentation by Rtnl1 in COS7 cells is more severe the longer Rtnl1 is over expressed. The discussion of GFP-Rtnl1 vs untagged Rtnl1 fission activity is useful for interpreting in vivo and in vitro results. And discussion of tubule diameter, Rtnl1 concentration, and fission as well as discussion of static versus force dynamics on tubules enhances the interpretation of the in vitro system.

We thank the reviewer for the positive assessment of our revised manuscript.

ER tubule fission is not well documented in the existing ER dynamics literature and, from my perspective, not generally accepted as a primary mechanism for ER network maintenance (usually overshadowed by tubule fusion, tubule sliding, and ring closure). The authors acknowledge this in the current manuscript, “The stochastic character of fission likely explains the low occurrence of fission events in the ER under normal circumstances when Rtnl1 concentration and pulling forces and speeds are within physiological limits.” However, tubule fission was recently shown by Guo et al 2018 Cell (mentioned twice by the authors in their rebuttal) and should be acknowledged in this manuscript. The findings by Espadas et al are notable and important for understanding of ER dynamics, morphology, and related disease. In response to their revisions and rebuttal, I suggest the following requirements before publication.

We apologize for the omission of the reference, as noted by the reviewer we used it in the rebuttal but it was meant to be included also in the manuscript.

1. ER tubule fission events are rare and under appreciated. ER tubule fission is not well known/accepted in the field of ER morphology. Please cite Guo et al 2018 Cell in the introduction. This would fit near the discussion of fragmentation and tubule fission. This specific paper is cited by Espadas et al in the revision rebuttal twice, since it contains examples of live mammalian ER tubule networks imaged at high speeds with specific instances of tubule fission. Citing another peer reviewed publication that used a different fluorescence imaging approach normalizes under appreciated ER fission events and elevates the current manuscript which offers a mechanism for these fission events.

Guo et al, Cell 2018 is now referred to in the introduction.

2. Please add some discussion of in vitro tubule diameter shown in Figure 4b and how it relates to physiological ER tubule diameters (physiological diameter measured by Shim et al 2012 PNAS, Terasaki 2018 J Cell Sci, Schroeder et al 2019

J Cell Bio). Interestingly, neuronal ER tubules are reported to have characteristically narrow ER tubules, which likely is relevant to rates of fission. The discussion is added after the description of the results shown in Fig. 4b.

3. I estimate their STED resolution is close to 150 nm, which is more in the domain of structured illumination (SIM) than standard STED implementation (40-80 nm resolution). I would recommend to them that they should use confocal imaging without heavy deconvolution for Fig2a and b, which I expect will produce results similar to that found using their STED images.

We do not exactly understand this recommendation. Fig. 2a-b was intended to show the emergence of bright fluorescent puncta in neuron cells under the indicated conditions. We note that similar puncta appeared in muscle cells (Fig. 2c) and also in cell culture (Fig. 3a, Supplementary Fig. 2), detected by confocal microscopy and epi-fluorescence. Hence, the emergence of the puncta is not specific to STED imaging. To further substantiate this point, we provide a snapshot (below) of neurons obtained by conventional spinning disk confocal microscopy (Andor Revolution) illustrating the difference between the control and Rtnl1-OE in neurons. The poor quality of these images also demonstrates why we used deconvolution to illustrate the phenotype. Nevertheless, the reviewer is absolutely right that heavy deconvolution is prone to various artifacts. To avoid those altogether, we performed the punctae area analysis de novo, using raw STED image stacks. As the reviewer correctly predicted, we obtained results qualitatively similar to those produced using deconvolved images. The results of this new analysis are now reported in new Fig. 2b that replaces the old version.

Rtnl1 OE

control

REVIEWERS' COMMENTS:

Reviewer #2 (Remarks to the Author):

The manuscript by Espadas et al now satisfies all my previously stated concerns. I recommend it for publication.